# How Do You Watch a Movie? HourHDVC: Hour-Long Hierarchical Dense Video Captioning

## Abstract

While existing Dense Video Captioning (DVC) research has shown promise for short video clips, current approaches struggle with hour-long videos due to a critical lack of datasets that capture long-term context and models capable of managing extensive temporal dependencies. To address these challenges, we introduce Hierarchical Dense Video Captioning (HDVC), a novel task designed for long-form videos that involves both scene-level and video-level narrative captioning. For this task, we propose HourHDVC, a new dataset providing comprehensive annotations for hour-long videos. We also present LOng COntext memory-based hierarchical dense video captioning (LOCO), an end-to-end model explicitly designed to manage extensive temporal dependencies by modeling the scene-to-narrative structure inherent in HDVC. LOCO leverages a two-tier memory system, Context-aware Memory and Long-term Context Memory, to maintain narrative coherence across extended durations. Experiments on HourHDVC demonstrate that LOCO establishes strong baselines for the HDVC task, while highlighting the remaining challenges of modeling long-form video narratives.

## 1 Introduction

As video content on platforms like YouTube and Netflix grows exponentially and videos reach longer durations, automatic fine-grained analysis of "*when*" and "*what*" events occur has become essential. This task is addressed by Dense Video Captioning (DVC) (Zhou et al., 2018b; Krishna et al., 2017; Wang et al., 2021; Yang et al., 2023b), which aims to detect event boundaries in untrimmed videos and generate descriptive captions for each event. It has shown promising results (Yang et al., 2023a; Kim et al., 2024), particularly for minute-level videos. Building on this progress, DVC shows significant potential to automate video analysis and alleviate laborious manual annotation; however, the community still lacks benchmarks that capture the complexity of hour-long videos, limiting systematic evaluation of models capable of reasoning over extended temporal context.

Applying DVC to hour-long videos presents two intertwined challenges. Existing DVC datasets primarily focus on short-duration content with concise annotations, insufficient to capture the causal and temporal relations in hour-long narratives. Meanwhile, prior DVC models struggle to maintain long-term contextual coherence and manage the extensive temporal dependencies required for narrative-level reasoning. To address these issues, we introduce a new long-form dataset and define the task of Hierarchical Dense Video Captioning (HDVC), which provides the foundation for studying DVC at the scale of hour-long videos. By constructing this benchmark, we establish a framework to investigate the systematic limitations of DVC in long-form settings, and we design a baseline model tailored to the nature of our dataset to show how these challenges can be addressed.

A central limitation lies in the mismatch between existing DVC datasets and the demands of long-form video understanding. Existing DVC datasets (Krishna et al., 2017; Zhou et al., 2018a; Huang et al., 2020) focus on short-duration content, spanning just a few minutes, and are annotated with concise sentences, often fewer than ten words per event. Such annotations prevent adequately representing real-world long-form videos such as movies, dramas, or hour-long daily videos, and they also hinder modeling of causal and temporal relations among events and implicit pragmatic cues

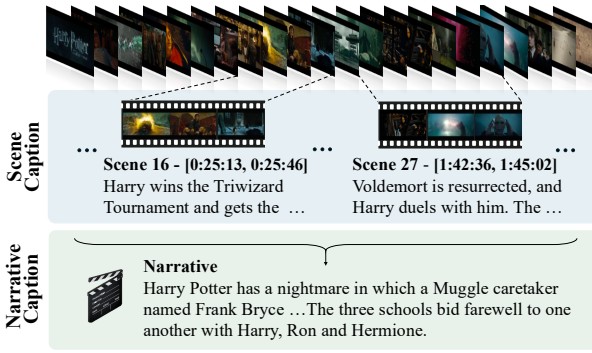
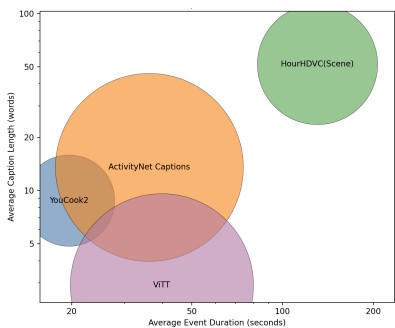

(a) Problem Definition of HDVC    (b) Comparison of Datasets in Event-level

Figure 1: **Task overview and dataset comparison.** We introduce Hierarchical Dense Video Captioning (HDVC), which generates scene-level captions and a single narrative for the entire video. As illustrated in (a), the objective is to segment scenes and provide a caption for each scene (scene-level) and a narrative for the entire video (narrative-level). As shown in (b), we compare DVC datasets at the event level: bubble size indicates the number of events, the x-axis denotes average event duration (seconds), and the y-axis denotes average caption length (words).

within narratives. The scarcity of datasets capturing extensive temporal context limits the development and evaluation of models for describing and understanding long-form videos.

To overcome these limitations, we introduce a new task, Hierarchical Dense Video Captioning (HDVC), and its corresponding large-scale dataset, HourHDVC, as illustrated in Fig. 1(a). The HDVC task requires models to perform two subtasks: (1) scene-level dense video captioning and (2) video-level narrative captioning. The hierarchical nature of this task is motivated by human cognition in building a coherent narrative context from continuous events (Zacks & Tversky, 2001). HourHDVC consists of hour-long videos comprehensively annotated for both HDVC subtasks. In this work, "context" denotes two complementary levels: (i) intra-scene context and (ii) inter-scene narrative context. The intra-scene context denotes the causal and temporal frameworks and the implicit intentions that structure actions and dialogues within a scene. The inter-scene narrative context denotes the cross-scene linkages and event progressions that structure and preserve the integrity of the global narrative over hour-long timelines.

Beyond dataset mismatch, existing approaches also face structural limitations when extended to hour-long videos, particularly in maintaining coherence and handling extensive temporal dependencies (Wang et al., 2021; Yang et al., 2023b; Kim et al., 2024). To illustrate these challenges and establish a solid baseline for future research, we present LOng COntext memory-based hierarchical dense video captioning (LOCO), explicitly designed for understanding hour-long videos. LOCO structurally integrates localized scene captioning and holistic narrative-level captioning within a unified architecture. To instantiate context in our model, we use a two-tier memory. The context-aware memory captures intra-scene context by aggregating temporally clustered visual events. In contrast, long-term context memory captures inter-scene narrative context by integrating these clusters with scene-level textual descriptions to maintain coherence over hour-long timelines. This design yields coherent paragraph-level outputs that retain local fidelity and global narrative structure. We also report scores from an LLM-based metric, $\text{ConSim}_{Llama3}$, to better account for the semantic relevance and coherence of paragraphs, as traditional n-gram metrics often miss these aspects.

Our main contributions can be summarized as follows:

- We introduce a new long-form dense video captioning task and present a corresponding hour-long dataset (i.e., **HourHDVC**). It contains hour-long videos annotated comprehensively for both scene-level dense video captioning and video-level narrative captioning tasks. Our approach is inspired by the hierarchical nature of human video understanding.

- We propose **LOCO**, an end-to-end model for hour-long videos that unifies scene localization, scene-level captioning, and narrative-level paragraph captioning within a single architecture. LOCO models the two-level context with a two-tier memory: a context-aware

memory to capture intra-scene context and a long-term memory to capture inter-scene narrative context. LOCO illustrates the utility of HourHDVC as a benchmark for long-form DVC, while serving as a strong baseline for future research on hour-long DVC.

- We conduct comprehensive experiments and analyses on the HourHDVC benchmark using LOCO and existing approaches. These results highlight the utility of HourHDVC and represent a first step toward unveiling the unexplored challenges of hour-long DVC.

## 2 RELATED WORK

### 2.1 VIDEO CAPTIONING DATASETS

Current video captioning datasets focus on short clips from 5 to 30 seconds in length (Chen & Dolan, 2011; Rohrbach et al., 2017; Wang et al., 2019; Xu et al., 2016; Rohrbach et al., 2015). Several datasets (Huang et al., 2020; Krishna et al., 2017; Zhou et al., 2018a) handle longer videos, typically averaging 1 to 5 minutes and sometimes extending beyond 20 minutes (Yang et al., 2023a). However, the captions in these datasets predominantly focus on short-term visual elements, such as atomic actions or the presence of specific objects. Several studies have incorporated hierarchical activity annotations for procedural videos (Bansal et al., 2022; Sener et al., 2022; Song et al., 2024; Tang et al., 2019; Zhukov et al., 2019). These datasets define a fixed taxonomy for activity labels at each hierarchical level and emphasize procedural activity recognition. Recent work (Islam et al., 2024) includes free-form natural language descriptions across multiple levels to capture the hierarchical structure of real-world long videos. This dataset focuses on visual information and abstracts the content into concise sentences, averaging approximately 15 words at the segment level and 25 at the video level. However, annotations in these datasets primarily describe simple visual actions or statuses, lacking rich contextual descriptions for long-term understanding.

In this study, we introduce a new dataset that advances prior work by leveraging both visual and speech information to construct context-based paragraph descriptions rich enough to capture the full context of a scene and the overarching narrative of the video, while uniquely combining three key characteristics absent in prior datasets: hour-long videos, paragraph-level annotations, and multi-level structure (Table 1). As shown in Fig. 1(b), our HourHDVC dataset contains scenes averaging 2 minutes in duration with detailed descriptions averaging 58 words, both comparatively long.

### 2.2 DENSE VIDEO CAPTIONING

DVC involves localizing events within untrimmed videos and generating descriptive captions for each detected event (Krishna et al., 2017). Traditional methods adopt a two-stage 'localize-then-describe' approach (Iashin & Rahtu, 2020a;b), handling event localization and captioning as separate tasks. Recent research addresses this limitation by developing end-to-end models that jointly learn both tasks, enhancing their interdependencies (Wang et al., 2021; Yang et al., 2023b; Zhou et al., 2024). $CM^2$ (Kim et al., 2024) and $HiCM^2$ (Kim et al., 2025) employ retrieval-augmented generation, retrieving relevant information from an external visual memory to provide additional context. Recent Multi-modal LLM (MLLM)-based methods (Ren et al., 2024; Huang et al., 2024; Lee et al., 2025) for dense video captioning, leverage the rich knowledge of LLMs to enhance semantic understanding and contextual coherence in captions. However, these methods are primarily designed for minute-long videos, which limits their use for hour-long videos.

To overcome these challenges, we introduce a unified end-to-end model that performs scene-to-narrative captioning for hour-long videos, where context-aware and long-term memories enable coherent understanding and contextual integration across extensive temporal spans.

## 3 HOURHDVC DATASET

We introduce a dataset for scene localization and description, with annotations of start/end times and textual descriptions. The dataset also includes hierarchically summarized narratives of entire videos. Scenes are divided into non-overlapping segments using semantic criteria such as location and episode changes, instead of relying only on visual transitions. Our approach requires long-form videos with dense visual and speech information, for which we leverage the Movie Audio

Table 1: **Comparison of video captioning datasets.** Hour-long denotes videos over one hour on average, Paragraph denotes multi-sentence annotations, and Multi-level denotes hierarchical annotations. A slash (/) is used for multi-level datasets to separate the word count per level.

| Dataset | Video Scale | | | Annotation Characteristics | | |
| --- | --- | --- | --- | --- | --- | --- |
| | #Videos | Avg. Duration (min) | Hour-Long | Avg. Words | Paragraph | Multi-level |
| ActivityNet Captions (Krishna et al., 2017) | 20K | 3 | ✗ | 13 | ✗ | ✗ |
| YouCook2 (Zhou et al., 2018a) | 2K | 6 | ✗ | 8 | ✗ | ✗ |
| ViTT (Huang et al., 2020) | 8K | 4 | ✗ | 2 | ✗ | ✗ |
| Youku Dense Caption (Zixuan Xiong, 2025) | 31K | 1 | ✗ | 17 | ✗ | ✗ |
| VidChapters-7M (Yang et al., 2023a) | 817K | 23 | ✗ | 5 | ✗ | ✗ |
| Ego4D-HCap (Islam et al., 2024) | 8K | 28 | ✗ | 7 / 15 / 25 | ✗ | ✓ |
| MAD (Soldan et al., 2022) | 650 | 110 | ✓ | 12 | ✗ | ✗ |
| MAD-V2 (Han et al., 2023) | 498 | 109 | ✓ | 9 | ✗ | ✗ |
| HourHDVC (Ours) | 498 | 109 | ✓ | 58 / 346 | ✓ | ✓ |

Description (MAD) dataset (Soldan et al., 2022) (Section 3.1). Because manual annotation for context-based segmentation and captioning is costly, we automatically generate annotations using LLMs grounded in high-quality human-authored visual descriptions to ensure reliability and quality (Section 3.2). Dataset statistics are reported in Section 3.3.

## 3.1 DATA COLLECTION

HourDVC is built upon the MAD-v2 (Han et al., 2023) dataset. The MAD (Soldan et al., 2022) and MAD-v2 offer valuable annotations for visual information through Audio Descriptions (AD). However, these datasets are primarily designed for aligning short clips, averaging about four seconds, with atomic actions or specific objects or persons. We extend the MAD-v2 dataset by localizing important scenes from videos and describing the scene based on long-term context, allowing a more comprehensive understanding of the narrative flow. Furthermore, the video-level narrative caption is annotated as a global context to understand the long-term context of the hour-long video.

## 3.2 DATA PROCESSING PIPELINE

HourHDVC's annotation uses a multi-stage pipeline leveraging LLMs. We begin by utilizing the dense annotations available in the MAD-v2 dataset, which include detailed speech transcripts and audio descriptions. To segment the movies into coherent scenes, we employed text-based scene segmentation using GPT-4o (Achiam et al., 2023).

**Generating Scene Segments and Captions.** In the initial stage of generating localized scene captions, we temporally segment scenes using the speech transcripts and visual descriptions. GPT-4o identifies points in the narrative where scene changes occur based on the speech transcript and audio descriptions. We give a prompt to GPT-4o as detecting a coherent context, such as a specific action or interaction sustained over an extended period, and ensuring that conversations based on a similar theme remain consistent at an abstract level, even if the detailed actions vary. This approach encourages GPT-4o to accurately delineate scene boundaries by focusing on context-based semantic criteria rather than relying on transitions between similar visual frames. The detail of the prompt is introduced in Supplementary Material. After temporal segmentation, we generate paragraph captions for each scene. Using GPT-4o, we produce concise captions that capture the events and contextual information within each segment, ensuring objectivity and preventing the inclusion of the LLM's subjective opinions. These scene-level captions also include timestamps for localization.

**Generating Video Narrative.** To generate the *video narrative*, we aggregate information from the scene captions and structure them according to the three-act structure: Setup, Confrontation, and Resolution. Using GPT-4o, we synthesize these scene captions to capture the exposition and introduction of conflict in the Setup phase, the rising action and climax in the Confrontation phase, and the falling action and denouement in the Resolution phase. This approach ensures that the video narrative caption provides a coherent overview of the entire movie, reflecting the overarching narrative, themes, and key plot points while maintaining contextual relevance to the scenes.

**Refining Evaluation Set.** To ensure the quality of our annotations, two domain experts perform human refinement on the evaluation data. These experts meticulously review the automatically gen-

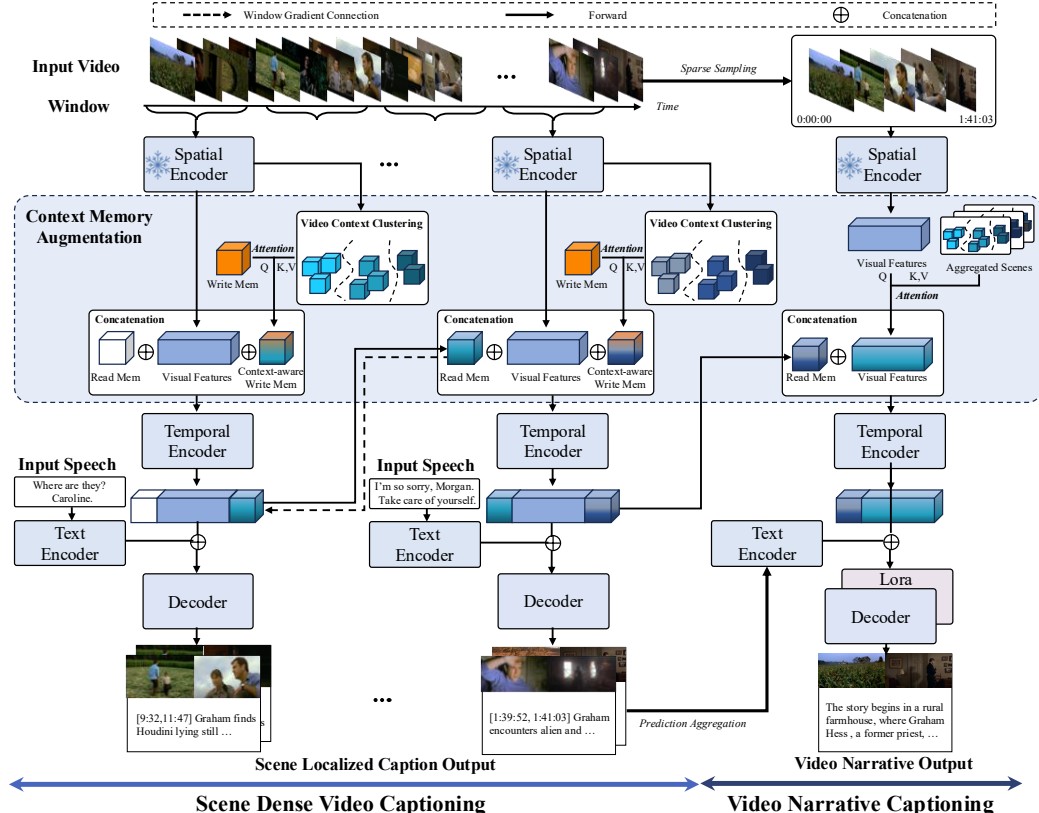

Figure 2: **Overview of the proposed framework.** In Scene Dense Video Captioning (left), a memory-augmented window processes video and speech. Inter-window connectivity is enhanced by passing information from context-aware write tokens to the read tokens of the next window. In Video Narrative Captioning (right), sparsely sampled frames are combined with a long-term context memory (aggregating predicted scene captions and visual contexts) to generate the final narrative. This scene-to-narrative structure is trained in an end-to-end manner.

erated scene segmentation and captions, making necessary adjustments to enhance accuracy and coherence. The refinement follows a two-stage expert verification procedure, where an initial pass corrects coarse timestamp or caption inconsistencies, and a second pass performs fine-grained adjustments. This human-in-the-loop process improves the reliability of the evaluation data providing a robust benchmark for accessing the model performance on long-form video understanding. Our method addresses scalability concerns by annotating whole videos with LLM and ensuring the quality of annotation with human refinement on the evaluation set.

## 3.3 DATA STATISTICS.

The HourHDVC dataset includes a diverse collection of movies with substantial variability in scene composition and narrative structure. On average, each scene spans 130.72 seconds (std 103.44), and each scene caption contains 58.06 words (std 42.53). Each movie has an average of 41.55 scenes (std 13.96), and scene captions include on average 3.54 sentences. This shows that the dataset effectively reflects the flexible scene lengths that follow the narrative flow of movies.For video narratives, each caption contains 346.74 words (std 62.15) and 17.28 sentences (std 3.67). These statistics demonstrate that HourHDVC provides comprehensive annotations at both scene and narrative levels, making it well-suited for hierarchical video captioning tasks.

# 4 METHOD

Given an hour-long video $X = \{x_t\}_{t=1}^{T}$ with aligned speech transcript $S = \{s_i\}_{i=1}^{N}$, the goal of HDVC is to generate (i) a set of scene-level captions $\hat{Y}^{scene} = \{\hat{y}_k^{scene}\}_{k=1}^{K}$ with timestamps, and (ii) a video-level narrative caption $\hat{Y}^{narr}$. Our framework (Fig. 2) jointly models both tasks in an end-to-end manner, explicitly capturing the scene-to-narrative structure. We denote short-term memory as $\mathcal{M}_{cont}$ and long-term memory as $\mathcal{M}_{long}$.

## 4.1 HIERARCHICAL SCENE-TO-NARRATIVE CAPTIONING

**Scene Dense Video Captioning.** We segment the video into windows $\{X^{(k)}\}_{k=1}^{K}$ with corresponding speech segments $S^{(k)}$. Each window is encoded by a spatial encoder $f_{sp}$, temporal encoder $f_{temp}$, and text encoder $f_{txt}$:

$$\mathbf{E}^{(k)} = f_{sp}(X^{(k)}), \quad \mathbf{u}^{(k)} = f_{txt}(S^{(k)}). \tag{1}$$

We denote the context memory for window $k$ as $\mathcal{M}_{cont}^{(k)}$, which stores the contextual representation computed from window $k$ (defined in detail in Section 4.2). The previous window's context memory is written as $\mathcal{M}_{cont}^{(k-1)}$. The temporal encoder integrates the previous memory, current visual features, and the current context memory:

$$\mathbf{h}^{(k)} = f_{temp}\big(\mathcal{M}_{cont}^{(k-1)}, \mathbf{E}^{(k)}, \mathcal{M}_{cont}^{(k)}\big), \tag{2}$$

where $\mathcal{M}_{cont}^{(0)}$ is initialized to zero for the first window.

The decoder $g_{scene}$ autoregressively generates localized scene captions:

$$\hat{y}_k^{scene} = g_{scene}(\mathbf{h}^{(k)}, \mathbf{u}^{(k)}). \tag{3}$$

**Video Narrative Captioning.** For the global task, sparsely sampled frames $X^{(g)}$ are combined with scene-level predictions $\hat{Y}^{scene}$ and long-term context memory $\mathcal{M}_{long}$. The narrative decoder $g_{narr}$ shares parameters with $g_{scene}$ and is fine-tuned with LoRA:

$$\hat{Y}^{narr} = g_{narr}(f_{temp}(f_{sp}(X^{(g)})), f_{txt}(\hat{Y}^{scene}), \mathcal{M}_{long}, \mathcal{M}_{cont}^{(k-1)}). \tag{4}$$

## 4.2 CONTEXT MEMORY

**Context-aware Memory.** For each window $k$, clustered visual features $\mathcal{C}^{(k)} = f_{cluster}(\mathbf{E}^{(k)})$ serve as key–value pairs. A learnable write token $\mathbf{q}_w$ attends to them to produce a raw context representation:

$$\tilde{\mathcal{M}}_{cont}^{(k)} = \text{Attn}(\mathbf{q}_w, \mathbf{K} = \mathcal{C}^{(k)}, \mathbf{V} = \mathcal{C}^{(k)}). \tag{5}$$

This is refined by the temporal encoder before being used in the next window:

$$\mathcal{M}_{cont}^{(k)} = f_{temp}(\tilde{\mathcal{M}}_{cont}^{(k)}). \tag{6}$$

**Long-term Context Memory.** To maintain narrative coherence, clustered context memories across windows are aggregated:

$$\mathcal{M}_{long} = \text{Aggregate}(\{\mathcal{M}_{cont}^{(k)}\}_{k=1}^{K}). \tag{7}$$

**Gradient Flow in Context-aware Memory.** Unlike conventional memory mechanisms that only pass forward features, our context-aware memory explicitly preserves gradient flow across windows. During training, the write token $\mathbf{q}_w$ not only stores contextual representations for the next window but also serves as a differentiable link, enabling gradients to propagate backward through memory updates. This design enhances inter-window connectivity, allowing early windows to refine later predictions while later windows retroactively influence the representation of earlier ones. Formally, for consecutive windows $k, k+1, k+2$, the shared write token updates context memories as

$$\mathcal{M}_{cont}^{(k+m)} = f_{temp}\big(\text{Attn}(\mathbf{q}_w, \mathcal{C}^{(k+m)})\big), \quad m \in \{0, 1, 2\}, \tag{8}$$

where $\mathbf{q}_w$ is optimized jointly across multiple windows. To balance efficiency with connectivity, we adopt gradient accumulation over three windows per training sample, ensuring that context is coherently modeled across long-form videos.

Table 2: **Performance comparison of scene dense video captioning.** SW denotes sliding window approaches. Bold means the highest score. Underline means 2nd score. Methods marked with an asterisk (*) are evaluated in zero-shot setting.

| Method | Captioning | | | | Localization |
|---|---|---|---|---|---|
| | $ConSim_{Llama3}$ | CIDEr | METEOR | Rouge-L | F1 |
| PDVC (Wang et al., 2021) | 10.41($\pm$0.22) | 0.27 | 1.37 | 4.52 | 5.83 |
| CM$^2$ (Kim et al., 2024) | 10.39($\pm$0.02) | 0.07 | 1.02 | 4.39 | 4.11 |
| Vid2Seq (Yang et al., 2023a) | 13.46($\pm$0.00) | 2.05 | 1.76 | 3.49 | 2.56 |
| TimeChat* (Ren et al., 2024) | 16.15($\pm$0.56) | 0.03 | 1.26 | 2.22 | 8.32 |
| VTimeLLM* (Huang et al., 2024) | 13.18($\pm$0.00) | 0.02 | 0.76 | 2.05 | 17.81 |
| ILCACM (Ge et al., 2025) | 11.42($\pm$0.06) | 0.10 | 1.43 | 3.03 | 17.55 |
| HiCM$^2$ (Kim et al., 2025) | 13.70($\pm$0.15) | **3.51** | 3.63 | 5.23 | 6.83 |
| PDVC-SW | 14.83($\pm$0.07) | 0.47 | 2.86 | 7.78 | **50.29** |
| Vid2Seq-SW | 18.49($\pm$0.12) | 2.27 | 2.87 | 4.88 | 40.16 |
| TimeChat-SW* | 15.82($\pm$0.29) | 0.17 | 1.28 | 2.35 | 30.39 |
| **LOCO (Ours)** | **33.04($\pm$0.23)** | 3.20 | **6.41** | **8.00** | 45.09 |

### 4.3 TRAINING OBJECTIVE

We jointly optimize scene and narrative losses using cross-entropy:

$$\mathcal{L}_{total} = \sum_{k=1}^{K} \mathcal{L}_{scene}(y_k^{scene}, \hat{y}_k^{scene}) + \mathcal{L}_{narr}(Y^{narr}, \hat{Y}^{narr}), \tag{9}$$

where $\mathcal{L}_{scene}$ and $\mathcal{L}_{narr}$ are token-level cross-entropy losses. LoRA parameters are updated only through $\mathcal{L}_{narr}$ to specialize the global decoder.

## 5 EXPERIMENTS

### 5.1 EXPERIMENTAL SETTINGS

All implementation details are provided in the supplementary material.

**Evaluation Metrics.** Using the official evaluation tool (Wang et al., 2020), we assess caption quality with CIDEr (Vedantam et al., 2015), BLEU4 (Papineni et al., 2002), METEOR (Banerjee & Lavie, 2005), and ROUGE-L (Lin, 2004). Scores are averaged over IoU thresholds 0.1, 0.3, 0.5, 0.7, reflecting the effect of temporal localization. For event localization, we report average precision, recall, and F1 (harmonic mean of precision and recall) over the same thresholds.

**ConSim: Context-Aware Evaluation Metric.** We introduce $ConSim_{Llama3}$, a context-aware evaluation metric for video paragraph captioning. $ConSim_{Llama3}$ leverages LLMs to assess contextual understanding. Unlike traditional N-gram metrics, which rely on word matching, by incorporating the semantic relevance and coherence of summaries. Therefore, $ConSim_{Llama3}$ provides a more semantically and contextually comprehensive evaluation. Existing N-gram-based metrics struggle to provide precise evaluations at the paragraph level. For scene dense video captioning, we evaluate ConSim at a timestamp IOU threshold of 0.3.

### 5.2 SCENE DENSE VIDEO CAPTIONING

In Table 2, we compare our method with prior approaches on HourHDVC. Our method achieves the best performance across three N-gram metrics and the LLM-based ConSim metric. Competing methods rely on sparse subsampling to reduce input size, which fails to capture fine-grained scene dynamics and long-term context. To address this, we also test their "-SW" variants, which segment videos into 10-minute clips processed independently. We applied this to three representative models (detection-based(Wang et al., 2021), sequence-based(Yang et al., 2023a), and MLLM-based(Ren et al., 2024)), and in all cases window-based processing outperforms sparse sampling. Moreover, our LOCO further surpasses them by explicitly modeling context with memory.

Table 3: **Performance of Video Narrative Captioning.**

| Method | $ConSim_{Llama3}$ | METEOR | BLEU4 |
|---|---|---|---|
| PDVC (Wang et al., 2021) | 12.00($\pm$0.81) | 5.96 | 0.00 |
| CM$^2$ (Kim et al., 2024) | 10.33($\pm$0.47) | 5.62 | 0.00 |
| Vid2Seq (Yang et al., 2023a) | 10.00($\pm$0.00) | 4.23 | 0.00 |
| MaMMut (Kuo et al., 2023) | 11.33($\pm$1.24) | 6.23 | 0.00 |
| VideoRecap (Islam et al., 2024) | 10.33($\pm$0.47) | 5.86 | 0.12 |
| V2Xum-LLM (Hua et al., 2025) | 10.00($\pm$0.00) | 3.24 | 0.01 |
| ILCACM (Ge et al., 2025) | 13.00($\pm$0.00) | 1.15 | 0.00 |
| HiCM$^2$ (Kim et al., 2025) | 16.00($\pm$0.00) | 6.57 | 0.19 |
| LOCO (Ours) | **18.66($\pm$1.24)** | **9.59** | **0.38** |

Table 4: **Ablation study to verify the effect of memory.**

| Context-aware Memory | Long-term Memory | Scene Dense Video Captioning | | | Video Narrative Captioning | |
|---|---|---|---|---|---|---|
| | | $ConSim_{Llama3}$ | METEOR | F1 | $ConSim_{Llama3}$ | METEOR |
| | | 30.29($\pm$0.27) | 6.32 | 43.96 | 12.66($\pm$0.47) | 7.70 |
| ✓ | | 30.21($\pm$0.22) | 6.11 | 40.24 | 13.66($\pm$0.47) | 7.74 |
| ✓ | ✓ | **33.04($\pm$0.23)** | **6.41** | **45.09** | **18.66($\pm$1.24)** | **9.59** |

## 5.3 VIDEO NARRATIVE CAPTIONING

In Table 3, we show the performance of video narrative captioning. Our method achieves higher scores compared with other methods which can be attributed to two key contributions. First, our model jointly learns scene and narrative captioning in an end-to-end manner, enabling knowledge sharing between the two subtasks. This facilitates the generation of more consistent and high-quality captions with a comprehensive understanding of the overall context. Second, our context-aware and long-term memory mechanism allows the model to capture rich information and long-range dependencies, which are crucial for understanding and captioning extended video content. By integrating these components, LOCO successfully produces informative and well-structured narrative captions.

## 5.4 ABLATION STUDY

**Ablation Study on the Effect of Memory.** To investigate the impact of different memory types on captioning performance, we conduct an ablation study, as shown in Table 4. The results demonstrate that incorporating context-aware memory already leads to video narrative captioning, as it helps retain important information across windows. The most significant performance boost is observed when long-term memory is also introduced, which enables the model to aggregate and leverage information from a broader temporal context. This leads to notable improvements in scene captioning metrics, including better fluency and coherence, as well as enhanced video narrative captioning, where the model achieves higher consistency and contextual awareness. These findings highlight the crucial role of long-term memory in refining captioning quality, ensuring that both subtasks are more comprehensive and contextually aligned.

**Ablation Study on Transcript Drop Robustness.** We study the model's sensitivity to transcript quality by simulating varying levels of ASR degradation using a random transcript drop mechanism. As shown in Figure 3, the model remains robust to moderate information loss. Performance is stable up to a 30% drop rate compared to the no-drop baseline. Beyond this point, performance gradually declines, with a notable drop once more than half of the transcript is removed and a severe degradation at a 90% drop rate. These results indicate that while high-quality transcripts are important for optimal performance, the model can tolerate moderate transcript sparsity without substantial harm, demonstrating resilience to realistic ASR imperfections.

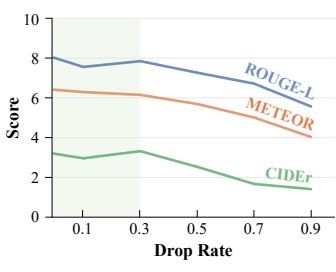 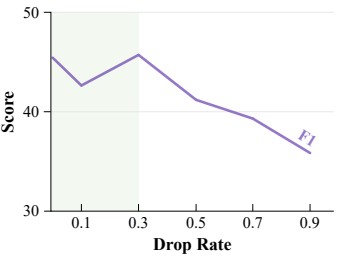

(a) Impact of Drop on Captioning      (b) Impact of Drop on Localization

Figure 3: **Robustness to Transcript Drop.** The shaded green region indicates that the model remains stable under small to moderate transcript drop rates.

Table 5: **Analysis of model generalizability on Ego4D-HCap dataset.**

| Method | | Captioning | | Localization | |
| --- | --- | --- | --- | --- | --- |
| | CIDEr | METEOR | Rouge-L | Recall | Precision |
| PDVC (Wang et al., 2021) | 162.08 | 19.93 | 32.85 | 21.57 | 52.81 |
| Vid2Seq (Yang et al., 2023a) | 145.29 | 19.49 | 38.71 | **53.20** | 73.32 |
| HiCM$^2$ (Kim et al., 2025) | 113.88 | 16.69 | 28.66 | 34.62 | 54.92 |
| **LOCO (Ours)** | **206.49** | **23.29** | **45.70** | 46.90 | **80.27** |

Table 6: **Analysis on timestamp noise of training dataset.**

| Training Dataset | Localization | |
| --- | --- | --- |
| | Recall | Precision |
| LLM Generated | 29.58($\pm$6.60) | 52.00($\pm$8.45) |
| Human Refined | 32.54($\pm$7.98) | 52.43($\pm$11.41) |

## 6 DISCUSSION

### 6.1 DATASET

**Quality Control.** To evaluate the quality of LLM-based annotations, we conducted a user study comparing captions generated by GPT with captions refined by two domain experts. Seven human evaluators rated the quality of captions for 20 video samples from each set, using a 10-point scale to assess how well they described the video. Results showed that the mean scores were nearly identical (8.49 for GPT-generated captions and 8.51 for expert-refined captions), indicating that, with well-designed data generation instructions, GPT-based annotations can reliably transform fine-grained MAD-v2 descriptions into high-quality annotations. To further examine the textual quality of GPT-generated annotations, we manually inspected all 364 scene-level descriptions used in the evaluation set. Among these GPT-generated captions, only 33 scenes (approximately 9%) required any textual refinement during expert review, and the hallucination rate was extremely low (about 1.9%). These observations confirm that the vast majority of GPT-generated descriptions were already accurate and factually grounded prior to refinement, reinforcing the overall reliability of our annotation pipeline.

**Reproducibility.** To assess reproducibility, we generated scene captions 10 times for the same video using our LLM-based annotation pipeline, and measured ConSim$_{Llama3}$ scores across all 45 pairs of captions to quantify consistency. The results show a high mean score of 85.92. These findings empirically support the reproducibility of annotation.

**Noise Analysis and Limitation.** We compared the timestamps of the LLM-generated scene captions for the ten evaluation movies with the ground truth scene timestamps, which are refined by domain experts. We observe an average IoU of 80.26. Therefore, it is expected that the annotation of training data might include approximately 20% noise in timestamps. The MAD dataset's audio descriptions were initially created as narration for visually impaired individuals, ensuring they do not overlap with actors' dialogues. Consequently, there are naturally discrepancies of a few seconds between the described scenes and actual scene boundaries. Also, additional noise arises from ASR transcription. The speech also contains instances where actors' utterances that are not audible are transcribed, and there is some noise where the speech start and end times do not precisely align. Note that these small noises are only included in the training data, and the evaluation set is clean (i.e., high-quality annotation) because two domain experts refine the annotations.

To further analyze the effect of such noise, we conducted controlled experiments using the same set of ten evaluation videos. For each video, we created sub-train/sub-eval splits (9 for training, 1 for evaluation), training models on either noisy (LLM-generated) or refined annotations. The trained

model was evaluated on the human-refined ground truth. As shown in Table 6, models trained on noisy annotations achieve recall of 29.58 (±6.60) and precision of 52.00 (±8.45), which are comparable to those trained on refined annotations (recall 32.54 (±7.98), precision 52.43 (±11.41)). These findings indicate that while refinement yields minor improvements, the influence of timestamp noise on model performance remains limited and does not pose a serious threat to the overall validity.

**Cross-Model Validation.** To test whether our annotations reflect a single model bias, we compared our GPT-4o captions with outputs from Gemini 2.5 Pro and GPT-5.1 for scenes with high temporal alignment (IoU $> 0.7$). Despite low lexical overlap (BLEU-4: 7.90 vs. Gemini, 5.20 vs. GPT-5.1), semantic agreement remained high (ConSim: 89.80 vs. Gemini, 87.40 vs. GPT-5.1). This pattern indicates that captions generated by different models convey the same underlying narrative facts, showing that our annotations are not driven by bias of a single LLM.

## 6.2 MODEL

**Generalizability of Model.** To validate the generalizability of LOCO, we analyzed the model on videos longer than one hour using the Ego4D-HCap dataset. We applied LOCO to the segment-level annotation task of Ego4d-HCap and compared its performance against established baselines, including PDVC, Vid2Seq, and HiCM[2]. As shown in Table 5, LOCO achieves the strongest overall captioning performance, with the highest CIDEr, METEOR, and Rouge-L scores among all competing methods. In localization, LOCO attains the highest precision and maintains competitive recall relative to Vid2Seq, outperforming PDVC and HiCM[2] in both metrics. These results indicate that LOCO produces richer and more accurate descriptions while preserving temporal consistency. LOCO demonstrates that its hierarchical design and context memory mechanism are not overfitted to HourHDVC but transfer effectively to other long-form video domains such as egocentric videos.

**Compute Efficiency.** We compare LOCO with Vid2Seq under an same sliding-window approach using the CLIP features. LOCO preserves nearly the same inference cost, requiring about 36 seconds to process a two-hour video compared to roughly 32 seconds for Vid2Seq-SW. In contrast, models without the sliding-window run much faster (5 seconds) but show severe degradation for long-video, as shown in Table 2. Overall, LOCO offers substantially stronger long-context performance while keeping computational cost comparable to Vid2Seq-SW, providing a favorable accuracy–efficiency trade-off for practical use. Note that the context-aware memory is only a single learnable token.

## 6.3 HUMAN ALIGNMENT OF CONSIM

To validate the contextual semantic similarity measure $\text{ConSim}_{Llama3}$, we have conducted a user study with 10 field experts, where we measure the Pearson correlation between human evaluation scores and automatic metrics. The seven randomly selected scene predictions and three randomly selected narrative predictions are used. Experimental results show that METEOR achieved a correlation of 0.27 and both ROUGE_L and BLEU_4 recorded –0.07. Whereas $\text{ConSim}_{Llama3}$ obtained a significantly higher correlation of 0.87. These findings indicate that $\text{ConSim}_{Llama3}$ aligns more closely with human judgment. To further assess the robustness of ConSim, we repeated this analysis with multiple LLM evaluators under the same evaluation prompt and instructions. In addition to $\text{ConSim}_{Llama3}$, $\text{ConSim}_{Llama3.3}$ and $\text{ConSim}_{Gpt5.1}$ achieved correlations of 0.75 and 0.84 with human score. This consistent pattern across different evaluators indicates that ConSim is not sensitive to the choice of LLM and reliably reflects human judgment.

## 7 CONCLUSION

In this study, we introduced HourHDVC, a new dataset designed for Hierarchical Dense Video Captioning, which provides comprehensive annotations for both scene-level localized captioning and video-level captioning of hour-long videos. To effectively model long-term dependencies, we proposed LOCO, an end-to-end framework that incorporates context-aware and long-term context memory mechanisms, improving context retention and coherence across windows. Experimental results on HourHDVC demonstrate that our approach effectively handles both scene localization and captioning in long-form videos, producing structured and contextually rich captions. Our work establishes a foundation for long-form dense video captioning, providing insights for future research in context-aware video understanding for hour-long videos.

## ETHICS STATEMENT

This research on long-form video captioning is conducted using publicly available datasets (MAD and MAD-v2), which already comply with privacy regulations. We acknowledge potential biases in automatically generated captions, such as gender or cultural stereotypes, and we have employed human refinement on the evaluation set to mitigate such risks. Furthermore, we are aware of possible misuse, for example, producing misleading or harmful content, and have taken steps to ensure the dataset and models are primarily oriented toward research in video understanding.

## REPRODUCIBILITY STATEMENT

We have made several efforts to ensure reproducibility. All key implementation details, including the architecture of LOCO, training procedures, and hyperparameter settings, are described in the main paper and supplementary material. The HourHDVC dataset construction pipeline, including LLM prompts and refinement process, is provided in detail in Section B of the appendix. Statistics of the dataset are reported in Section 3.3, and additional analyses are presented in the appendix. We also describe hardware configurations and training requirements to support reproducibility. Code, model weights, and processed dataset splits will be released with the paper to facilitate replication of our results.

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

# Supplementary Material

## A IMPLEMENTATION DETAILS.

We set the window size to 600 frames extracted by 1fps and the window stride to 600, resulting in no overlapping. Video frames are extracted at a rate of one frame per second. The sequences are then either subsampled or padded to ensure a total of $F$ frames, with $F$ set to 100 for each window. Both the text encoder and decoder are initialized using the pre-trained T5-Base model (Raffel et al., 2020). All experiments are conducted on four RTX 4090 GPUs. Our baseline model is built on the sequence-to-sequence architecture of the Vid2Seq model (Yang et al., 2023b), which is trained on VidChapters (Yang et al., 2023a). Our model has 315M trainable parameters. For the video narrative captioning, we account for potential noise in scene predictions during inference. To enhance robustness, we add word-level noise by randomly perturbing 50-80% of the words in the scene prediction which is used as input for video narrative captioning during training.

## B ADDITIONAL ANALYSIS OF HOURHDVC

**Pipeline of Data Generation of HourHDVC.** Figure 4 shows the pipeline used for data generation in HourHDVC. As shown in the figure, we input densely annotated audio descriptions and complementary speech data into a Large Language Model (LLM) to perform text-based scene segmentation and captioning. This process yields captions and localized timestamps for the scenes. For the evaluation set, these outputs are further refined by two annotators. Subsequently, by providing the scene captions to the LLM, we generate a video narrative structured according to the three-act framework of setup, confrontation, and resolution. Through text-based scene segmentation and captioning, we obtain scenes that effectively consider semantic context.

**Instruction for Generation of HourHDVC.** In Algorithm 1, we present the instructions that are provided to the LLM as a prompt for scene segmentation and captioning in our HourHDVC generation. Initially, we outline the characteristics of the input data as the first guide. We inform that the Audio Description is a visual description of a short video that may include slight temporal noise, and we also inform the potential noise and contextual information in the speech transcript extracted by the Automatic Speech Recognition (ASR) model WhisperX. Furthermore, we specify seven instructions, including the definition of a scene, guidelines for localization using noisy input, maintaining objectivity and conciseness, and preventing redundant generation. Finally, by separating the scene segmentation form and the scene captioning form, we encourage the model to proceed step-by-step.

We also present the instructions provided to the LLM for video narrative captioning in our HourHDVC generation. As shown in Algorithm 2, we outline the information of the input scene captioning consisting of paragraphs. Then, we guide the generation of a narrative captioning structured according to the three-act framework of setup, confrontation, and resolution, based on four instructions. By providing detailed and appropriate guidance, we obtain outputs with contextually well-considered captions and minimal noise in both the scene and video levels.

**Comparision with Naive Instruction for Generation of HourHDVC.** In Algorithm 3, we present the naive instruction for data generation. We provide a brief explanation of the tasks the LLM should perform and the format of the output. In Figure 8, we show the LLM scene segmentation and captioning outputs generated using both the naive instruction and our instruction. The naive instruction tends to produce localized scene timestamps that do not align with the generated captions, showing significant errors of 437 seconds in the start time and 245 seconds in the end time compared to human localization. In contrast, our detailed instruction yields localized timestamps that are well-aligned with the generated captions compared to human localization. With a start time difference of 20 seconds and an end time difference of 5 seconds, this demonstrates the effectiveness of our well-defined instruction in text-based scene segmentation.

**Scene vs. Narrative Scatter.** Figure 5 illustrates the relationship between duration and caption length at both the scene and narrative levels. On the left, the scene scatter shows that most scenes fall within 20–120 seconds, with captions averaging around 40–60 words, though some extend beyond

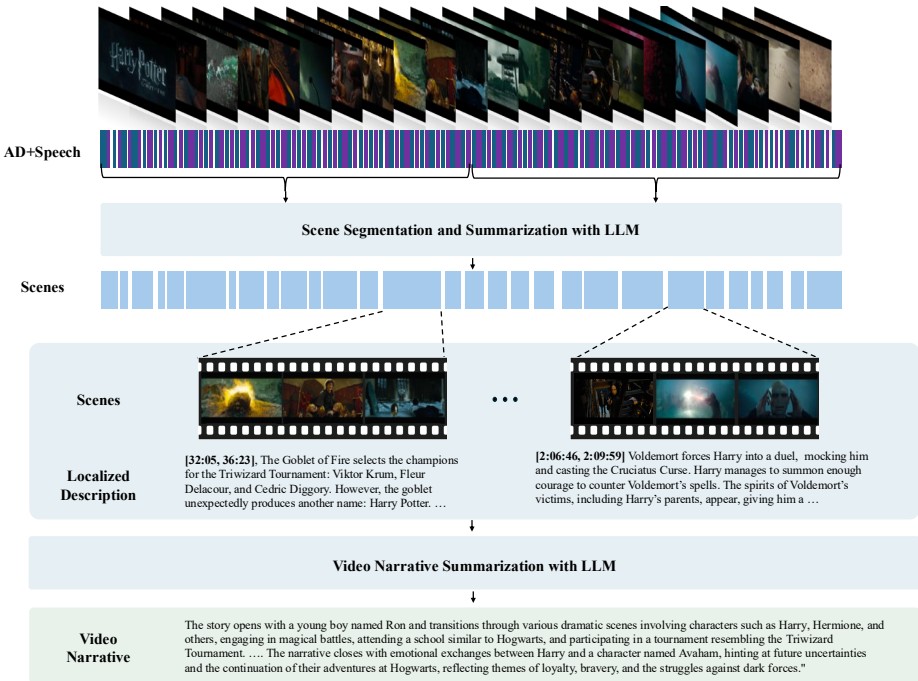

Figure 4: **Pipeline of Data Generation of HourHDVC.**

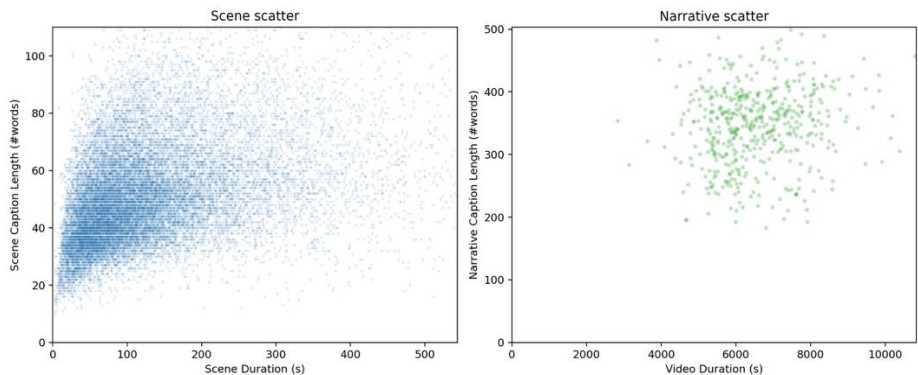

Figure 5: **Scatter analysis of HourHDVC.** Scatter plots of scene duration vs. caption length and video duration vs. narrative caption length.

300 seconds and 100 words. This distribution indicates that scene-level annotations are not confined to short actions but cover diverse temporal spans with sufficiently detailed descriptions. On the right, the narrative scatter demonstrates that despite varying video durations (ranging from about 1 to 3 hours), narrative captions are consistently long, concentrated between 250 and 400 words. This suggests that narrative annotations provide comprehensive story-level summaries, relatively independent of raw video length. Together, these distributions validate that HourHDVC captures both fine-grained scene understanding and coherent long-form narrative abstraction.

**Dataset Distribution.** Figure 6 illustrates the statistical characteristics of HourHDVC at both the video and scene levels. At the video level, most videos range between 5,000–8,000 seconds (approximately 1.5–2.2 hours), confirming that the dataset primarily consists of hour-long movies. The narrative captions exhibit a relatively narrow distribution, concentrated around 300–400 words, which ensures consistency in the amount of contextual detail provided across videos regardless of duration. At the scene level, the distribution of scene durations is heavily skewed toward shorter spans, with the majority lasting under 150 seconds. Nevertheless, the long tail extends beyond 500 seconds,

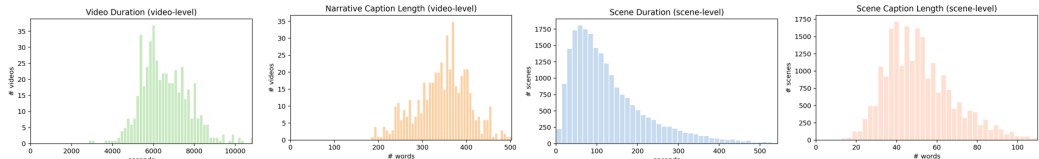

Figure 6: **Data analysis of HourHDVC.** Distribution of video duration, narrative caption length, scene duration, and scene caption length.

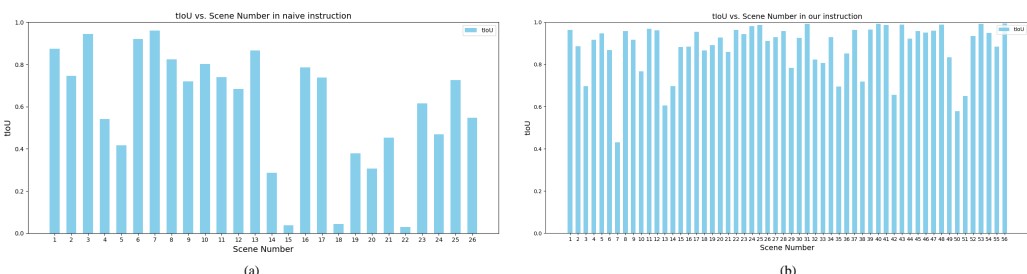

Figure 7: (a) IOU between human annotation and temporal localization by GPT4o with naive instruction from "Harry Potter and the Goblet of Fire" movie (Average IoU of 59.45). The domain expert refines timestamps corresponding to the captions. (b) IOU between temporal localization by human annotation and GPT4o with instruction used in this study (Average IoU of 87.87).

reflecting the diversity of narrative pacing across different movies. The corresponding scene captions cluster around 40–60 words, indicating that each scene is described with sufficient granularity to capture complex interactions and context beyond simple atomic actions. Together, these distributions highlight that HourHDVC balances long-form narrative coherence with detailed local scene descriptions, making it a unique benchmark for hierarchical dense video captioning.

**Effect of Guided Instruction for Text-based Scene Captioning.** In this study, we generate data for scene segmentation and captioning using LLMs. Figure 7 shows the comparison of scene segmentation and captioning performed with naive instructions against our well-defined instructions.

## C  DATA LEAKAGE

As illustrated in Figure 9, although GPT-4o and Llama3-70B are not provided with any information regarding "Signs", one of the evaluation sets in our HourHDVC dataset, they can still convey information about the movie. For example, GPT-4o knows the director's name and plot details, while Llama3-70B possesses knowledge not only of the plot but also of the movie review. This may be because these data are already used to train large language models (LLMs) such as GPT-4o and Llama3-70B, resulting in these models already possessing knowledge of the movies. This indicates that data leakage occurs, meaning we cannot fairly evaluate the video narrative captioning models that utilize GPT-4o or Llama3 within our HourHDVC dataset. For example, in our narrative captioning task, GPT-4o using Oracle scene captions as input achieves evaluation scores of $ConSim_{Llama3}$ 87.33, METEOR 13.84, and ROUGE-L 20.38. Similarly, Llama-70B using Oracle scene captions as input attains $ConSim_{Llama3}$ 78.99, METEOR 17.71, and ROUGE-L 24.91. Therefore, in this study, we do not use LLMs for video narrative captioning. This issue is specific to the video narrative captioning task, as scene dense video captioning is grounded in localizing and describing visual events rather than recalling plot information.

## D  ADDITIONAL ANALYSIS OF LOCO

**Ablation Study on the Window Size.** We evaluate four window sizes to analyze their effect on temporal modeling. As shown in Table 7, a 600-second window provides the best overall balance

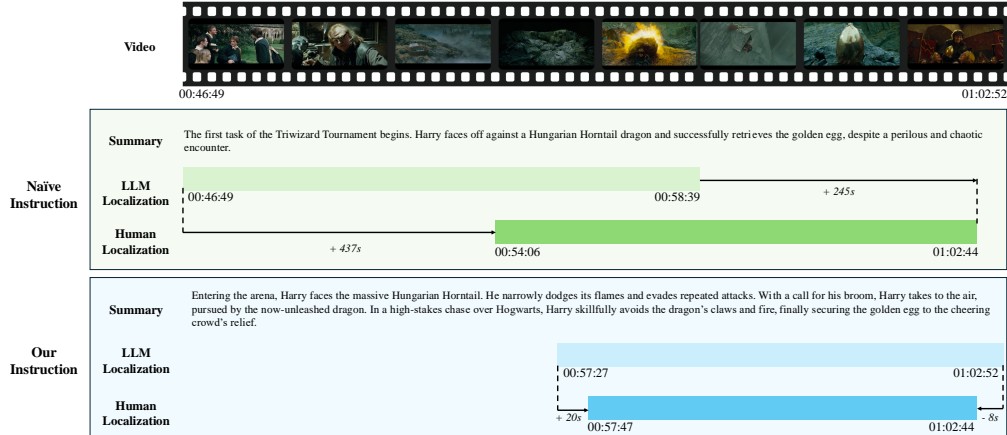

Figure 8: **Comparison of our Instruction with Naive Instruction for Generation of HourHDVC**

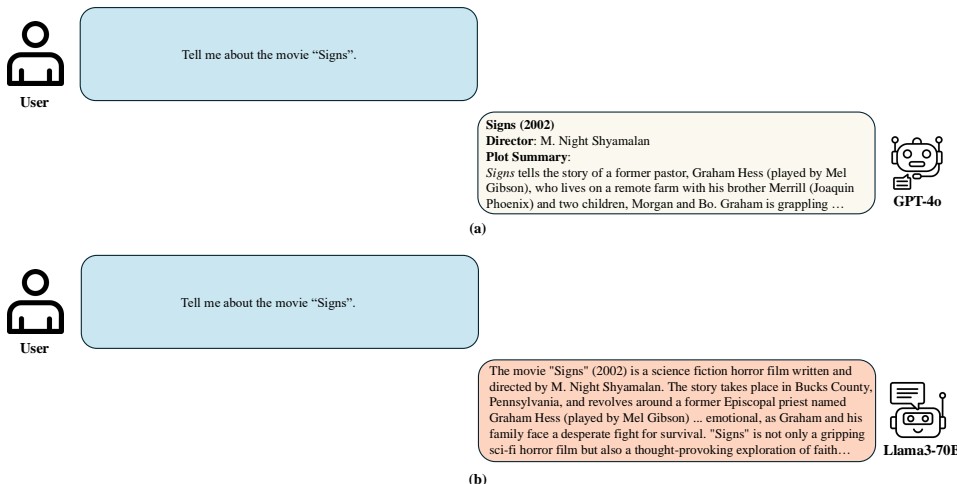

Figure 9: **Data Leakage in LLM Training Data.** In Figure (a), we query GPT-4o about "Signs", which is one of the movies in the evaluation set of HourHDVC. In Figure (b), we query Llama 3-70B about the same movie. Both models know about the "Signs". This indicates that data leakage has occurred, as information about our video narrative captioning evaluation set videos is present in the training data of the LLMs.

Table 7: **Ablation study on window size.**

| Window Size (seconds) | Scene Dense Video Captioning | | | Video Narrative Captioning | |
|---|---|---|---|---|---|
| | METEOR | ROUGE_L | F1 | METEOR | BLEU4 |
| 300 | 6.72 | 7.86 | 42.68 | 9.07 | 0.17 |
| **600(Ours)** | 6.41 | 8.00 | 45.09 | 9.59 | 0.38 |
| 900 | 6.28 | 6.70 | 30.22 | 8.37 | 0.32 |
| 1200 | 6.17 | 8.38 | 29.95 | 9.12 | 0.55 |

for both Scene Dense Video Captioning and Video Narrative Captioning. Smaller windows (300s) encompass multiple scenes because the average scene lasts only 131 seconds, causing scene boundaries to be split across windows and reducing localization accuracy. In contrast, very large windows (900–1200s) lead to low frame-sampling density within each window, making it difficult to capture short but visually important events. Overall, 600 seconds offers a stable trade-off between preserving local detail and capturing long-range context, and is adopted as our default configuration.

Table 8: **Ablation study to verify the effect of Context Memory.**

| Lora | Context Memory | Scene Dense Video Captioning | | | Video Narrative Captioning | |
|---|---|---|---|---|---|---|
| | | ConSim$_{Llama3}$ | METEOR | F1 | ConSim$_{Llama3}$ | METEOR |
| | | 29.33($\pm$0.27) | 5.98 | 38.53 | 12.33($\pm$1.24) | 8.20 |
| | ✓ | 32.88($\pm$0.20) | 6.38 | 42.82 | **19.33($\pm$1.69)** | 9.56 |
| ✓ | | 30.29($\pm$0.27) | 6.32 | 43.96 | 12.66($\pm$0.47) | 7.70 |
| ✓ | ✓ | **33.04($\pm$0.23)** | **6.41** | **45.09** | 18.66($\pm$1.24) | **9.59** |

Table 9: **Effect of context-aware memory type on captioning.** Memory Only refers to using written tokens without visual attention. Frame-aware Memory denotes attending to both input frames and write memory. Scene-aware Memory (Our method) clusters context memories based on scene structures.

| Context-aware Memory Type | Scene Dense Video Captioning | | | Video Narrative Captioning | |
|---|---|---|---|---|---|
| | ConSim$_{Llama3}$ | METEOR | F1 | ConSim$_{Llama3}$ | METEOR |
| Memory Only | 30.94($\pm$1.79) | 6.05 | 42.76 | **19.00($\pm$1.41)** | 9.56 |
| Frame-aware Memory | 31.67 ($\pm$0.14) | 6.35 | 42.60 | 17.99($\pm$1.63) | 9.46 |
| Scene-aware Memory | **33.04($\pm$0.23)** | **6.41** | **45.09** | 18.66($\pm$1.24) | **9.59** |

Table 10: **Impact of context-aware memory size.** Bold means the highest score.

| Memory Size | Local Summarization | | | Global Summarization | |
|---|---|---|---|---|---|
| | SemSim$_{Llama3}$ | METEOR | F1 | SemSim$_{Llama3}$ | METEOR |
| 0 | 30.29($\pm$0.27) | 6.32 | 43.96 | 12.66($\pm$0.47) | 7.70 |
| 1 | **33.04($\pm$0.23)** | **6.41** | **45.09** | **18.66($\pm$1.24)** | **9.59** |
| 2 | 30.88($\pm$0.42) | 5.86 | 42.65 | 17.00($\pm$1.41) | 9.08 |
| 3 | 31.09($\pm$0.29) | 5.73 | 39.83 | 17.66($\pm$0.47$-$) | 9.19 |

Table 11: **Effect of long-term memory type.** Window scene memory denotes clustered visual contexts from the scene captioning stage, while Window scene prediction denotes the scene time-caption predictions.

| Window Scene Memory | Window Scene Prediction | Scene Dense Video Captioning | | | Video Narrative Captioning | |
|---|---|---|---|---|---|---|
| | | $ConSim_{Llama3}$ | METEOR | F1 | $ConSim_{Llama3}$ | METEOR |
| | | 30.21($\pm$0.22) | 6.11 | 40.24 | 13.66($\pm$0.47) | 7.74 |
| ✓ | | 30.90($\pm$0.23) | 5.92 | 40.31 | 14.33($\pm$1.24) | 8.29 |
| | ✓ | 30.16($\pm$0.27) | 6.51 | 44.70 | 14.33($\pm$0.47) | 9.18 |
| ✓ | ✓ | **33.04($\pm$0.23)** | **6.41** | **45.09** | **18.66($\pm$1.24)** | **9.59** |

**Compute Efficiency and Practical Feasibility.** Although LOCO incorporates hierarchical modeling and two-tier context memory, the method remains computationally practical for long-form video. Under identical sliding-window settings and using the same pre-extracted CLIP features, LOCO matches the runtime profile of Vid2Seq-SW, requiring roughly 36 seconds to process a two-hour video compared to 32 seconds for Vid2Seq-SW. Notably, methods that avoid sliding-window processing run faster but suffer severe accuracy degradation (e.g., ConSim drops from 33.04 to 13.46), indicating that efficient long-context modeling—not sparse sampling—is the key computational bottleneck. Importantly, LOCO achieves substantially higher accuracy (ConSim 33.04 vs. 18.49 for Vid2Seq-SW) at comparable cost, highlighting that its hierarchical design improves long-range reasoning without incurring meaningful overhead. These findings suggest that LOCO offers a favorable accuracy–efficiency trade-off that makes it feasible for real-world long-video applications.

**Analysis of Robustness to Noisy Speech.** We examine the robustness of LOCO to noisy or music-heavy audio environments by comparing two contrasting movies from the evaluation set: Les Misérables, which contains frequent songs and dense background music with nearly half of all

speech overlapped, and Signs, which features comparatively clean speech with minimal overlap. We evaluate captioning performance under both conditions using metrics averaged over four tIoU thresholds (0.1/0.3/0.5/0.7). As expected, the musical setting of Les Misérables leads to a slight degradation in metrics such as METEOR and CIDEr relative to Signs, reflecting the challenges posed by heavily masked dialogue. Nevertheless, the model maintains stable captioning quality without catastrophic failure, indicating that LOCO effectively leverages visual cues to compensate for noisy or sparse speech. This suggests that while clean transcripts improve performance, the model remains resilient to significant audio corruption.

**Analysis on Long-Term Memory Type.** To analyze the effect of long-term context memory type, we conduct an ablation study, as shown in Table 11. Experimental results show that window scene memory alone already improves captioning performance by enhancing contextual retention across scene and narrative windows. When window scene prediction is incorporated, further improvements are observed, particularly in scene captioning.The combination of window scene memory and window scene prediction achieves the best performance across all evaluation metrics, indicating that integrating predictive mechanisms allows the model to anticipate and structure long-range dependencies more effectively. These findings highlight that long-term context memory plays a crucial role in improving both scene coherence and narrative consistency, ultimately leading to more accurate and contextually aligned captioning.

**Ablation Study on Context Memory.** To evaluate the effect of Context Memory, we conduct an ablation study, as shown in Table 8. The results indicate that Context Memory significantly enhances captioning performance. The Context Memory enables the model to generate fluency and relevance descriptions, while also yielding more precise segment boundaries. Additionally, video narrative captioning benefits from increased coherence and contextual consistency, achieving the best overall performance. When we utilize LoRA, scene captioning improves as LoRA stabilizes scene optimization by reducing interference from the narrative captioning objective. When Context Memory is combined with LoRA, scene captioning performance can be further improved.

These findings confirm that integrating Context Memory allows the model to generate more contextually rich and structurally coherent captions, playing a crucial role in both scene dense video captioning and video narrative captioning tasks.

**Effect of Context-aware Memory Type on Captioning.** Table 9 compares different context-aware memory types for captioning tasks. The results indicate that Scene-aware Memory, our method, achieves the best overall performance across both subtasks. Compared to Memory Only, which writes tokens to memory without visual attention, both Frame-aware Memory and Scene-aware Memory show improvements, demonstrating the importance of incorporating visual cues into memory mechanisms. Among them, Scene-aware Memory yields the highest scene dense video captioning metrics, suggesting that leveraging structured scene-based memory enhances coherence and fluency. These findings highlight the effectiveness of our clustering-based Scene-aware Memory, which enables more structured long-term context modeling, resulting in improved captioning quality.

**Impact of Context-aware Memory Size on Hierarchical Dense Video Captioning.** Table 10 presents the impact of context-aware memory size on captioning performance. The results indicate that incorporating a moderate amount of memory (size = 1) leads to the most significant improvements in both scene dense video captioning and video narrative captioning, achieving the highest fluency, coherence, and contextual accuracy. While increasing the memory size beyond this point still contributes to narrative captioning performance, the gains become marginal, and scene captioning metrics, such as METEOR and F1, begin to degrade. This suggests that excessive memory may introduce redundant or less relevant contextual information, making it harder for the model to focus on essential details. Therefore, a balanced approach to memory size selection is crucial for maximizing both scene dense video captioning and video narrative captioning effectiveness.

**Impact of Gradient Flow on Memory Connection and Captioning.** Table 12 presents the impact of memory gradient connection on captioning performance. The results indicate that enabling gradient flow across inter-window memory leads to moderate improvements in scene dense video captioning, particularly in terms of coherence and consistency, as reflected in the higher F1 and METEOR scores. While no gradient flow achieves a slightly higher LLM-Sim score in video narrative captioning, it underperforms in other key metrics, suggesting that preventing gradient propagation

Table 12: **Ablation study to verify the effect of memory gradient connection.** No Gradient Flow refers to blocking gradient propagation through inter-window memory. Gradient Flow denotes allowing gradients to flow through memory connections to refine representations across windows. Bold means the highest score.

| Inter-window Connection | Scene Dense Video Captioning | | | Video Narrative Captioning | |
|---|---|---|---|---|---|
| | $\text{ConSim}_{Llama3}$ | METEOR | F1 | $\text{ConSim}_{Llama3}$ | METEOR |
| No Gradient Flow | 31.61($\pm$0.28) | 6.57 | 41.55 | 19.33($\pm$0.47) | 9.55 |
| Gradient Flow | 33.04($\pm$0.23) | 6.41 | 45.09 | 18.66($\pm$1.24) | 9.59 |

Table 13: **Noise addition test to show the needs for $\text{ConSim}_{Llama3}$.**

| Noise Ratio | $\text{ConSim}_{Llama3}$ | METEOR | BLEU4 |
|---|---|---|---|
| 0%(Oracle) | 100 | 100 | 100 |
| 10% | 99.5 | 52.0 | 74.0 |
| 20% | 95.5 | 42.0 | 54.7 |
| 30% | 86.1 | 34.6 | 39.4 |
| 40% | 67.2 | 28.3 | 26.5 |
| 50% | 41.5 | 22.8 | 17.0 |

may reduce the model's ability to leverage long-range dependencies. These results indicate that gradient flow can be beneficial for enhancing representation learning across windows, though its overall impact depends on the trade-off between scene consistency and video narrative contextual alignment.

**Example of Context-Aware Clustering.** To verify the behavior of our context-aware memory, we visualize its clustering results, as shown in Figure 11. The figure illustrates how frames are assigned to different clusters, reflecting the model's grouping of visually and semantically related segments. This serves as a sanity check to confirm that our method organizes frames in a way that aligns with scene structures, ensuring that the memory mechanism retains meaningful contextual information for captioning.

# E    ADDITIONAL ANALYSIS OF CONSIM

**Instruction for evaluating with $\text{ConSim}_{Llama3}$.** In Algorithm 4 and Algorithm 5, we present the instructions used for measuring $\text{ConSim}_{Llama3}$. Here, we ensure the ability of LLM to evaluate contextual semantic similarity through two procedures. First, we provide detailed guidance by proposing specific evaluation criteria, such as exploring semantic information based on scene understanding, conditions for person matching, and considerations with evaluation methods for contextual semantic similarity and temporal alignment. Second, we induce step-by-step reasoning through a chain of thought. By illustrating the reasoning process in a given example, we enable the LLM to accurately assess the two evaluation criteria. By repeating this process three times and averaging the scores of the two criteria obtained, we achieve a relatively accurate semantic evaluation score.

In the scene dense video captioning of HourHDVC, since we obtain captions that include localization, temporal alignment should be considered. Therefore, we assess temporal alignment during the scene dense video captioning but do not evaluate it in the video narrative captioning stage.

**Comparison our $\text{ConSim}_{Llama3}$ with Traditional N-gram based Metrics.** We present both qualitative and quantitative comparisons between our $\text{ConSim}_{Llama3}$ and METEOR that is the traditional N-gram-based metric. To further demonstrate the robustness of our proposed $\text{ConSim}_{Llama3}$ metric, we conducted two complementary noise addition tests at different granularities—one at the sentence level and the other at the word level. In the sentence-level test (Figure 10), we began with an oracle state where the predictions and ground truths were identical for all 10 movies. We then progressively replaced $n\%$ of the sentences in each movie with sentences from different movies, gradually transitioning toward a random state. Our $\text{ConSim}_{Llama3}$ metric exhibited a stable decline in evaluation

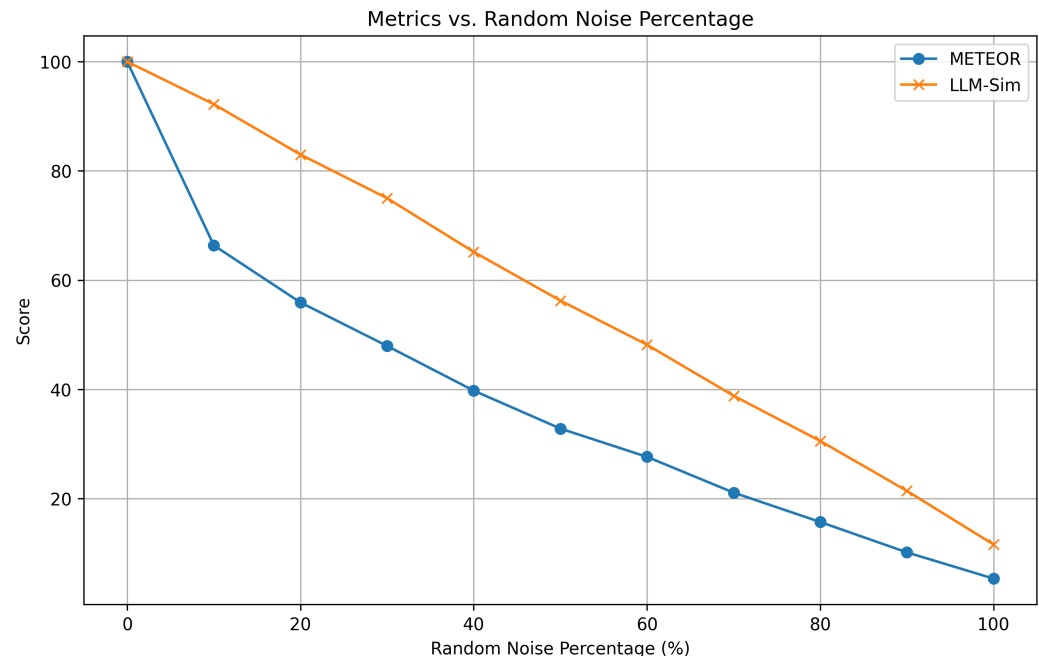

Figure 10: **Noise addition test in sentence level to show the needs for ConSim$_{Llama3}$.**

scores proportional to the degree of noise insertion, whereas the n-gram-based metric METEOR was highly sensitive, dropping significantly to approximately 66 points even with just 10% noise.

Similarly, in the word-level test (Table 13), an oracle scenario was first established where the scene dense video captioning for the ten movies exactly matched the ground truth. The noise was incrementally introduced by replacing 10% of the words with random words drawn from the entire vocabulary of caption words. Consistent with the sentence-level findings, our ConSim$_{Llama3}$ metric showed a stable decrease in evaluation scores with noise insertion, while METEOR suffered a substantial drop, reaching approximately 52 points at 10% noise. These results collectively demonstrate that ConSim$_{Llama3}$ reliably measures contextual semantic similarity between predictions and ground truth, exhibiting robustness and consistency across different levels of textual perturbations. We also show the qualitative comparison results between our ConSim$_{Llama3}$ and N-gram based metrics in Figures 12, 13, 14, and 15. In these figures, we compare two semantically identical sentences using both N-gram-based metrics and our ConSim$_{Llama3}$. Traditional N-gram-based metrics have difficulty making meaningful evaluations for expressions that use different words but convey the same meaning. Specifically, as shown in Figure 12, when there are no consecutive identical words between the prediction and the ground truth, N-gram-based metrics fail to measure contextual semantic similarity, as evidenced by a BLEU4 score of 0.08. In contrast, our ConSim$_{Llama3}$ demonstrates that it can perform sentence similarity evaluations at the semantic level more flexibly and accurately, free from the constraints of identical words.

## F    QUALITATIVE RESULTS

**Examples of HourHDVC** In Figure 16, we present examples from the HourHDVC training set. It can be observed that even the training set without human refinement exhibits quite high localization accuracy and rich expressions. In Figure 17, we show examples from the HourHDVC evaluation set. In both the training and evaluation sets, we can observe scene segmentation and captioning that thoughtfully consider context, and we can confirm a narrative caption that contextually describes the overall content effectively.

**Prediction Example of Ours.** In Figure 18, we present examples from our HourHDVC dataset alongside predictions generated by our model. The scene captions effectively incorporate context

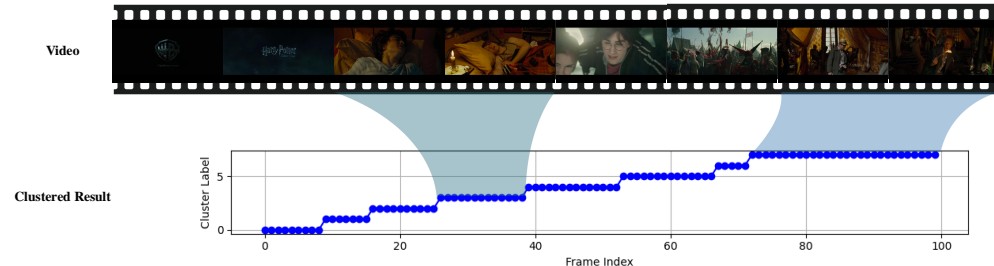

Figure 11: **Visualization of the clustered results in context-aware memory.** The upper section shows sampled frames from the video, while the lower section presents the corresponding cluster assignments. This visualization serves as a sanity check to confirm that the context-aware memory groups semantically related frames, aligning with scene structures.

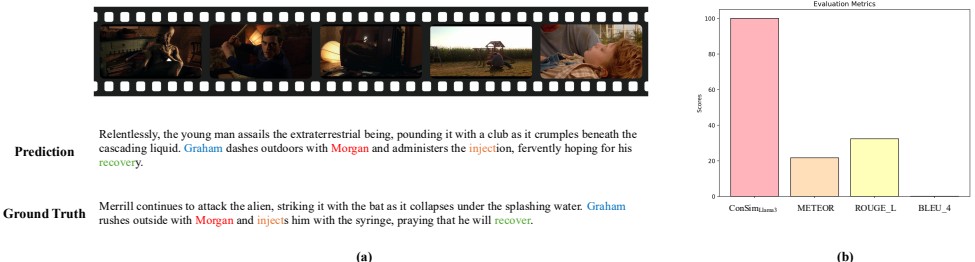

Figure 12: **Comparison of our ConSim$_{Llama3}$ with N-gram based metrics.** In (a), we compare two semantically identical sentences using both N-gram-based metrics and our ConSim$_{Llama3}$. In the two sentences, words highlighted in the same color represent identical words. We normalized all metrics to a uniform range of 0 to 100. In this comparison, ConSim$_{Llama3}$ measures a score of 100.00, while METEOR, Rouge-L, and BLEU4 scored 21.67, 32.35, and 0.08, respectively, in (b). These results show that our ConSim$_{Llama3}$ goes beyond simple word matching and can evaluate contextual semantic similarity.

through visual descriptions and speech. These results demonstrate that our method successfully performs context-aware scene dense video captioning. However, in the case of the video narrative captioning, we generate a video narrative caption that repeats certain sentences and expressions. Due to the conditions of sparsely sampled video frames and the model's sensitivity to scene captions, these remain challenging aspects that indicate the need for further improvement in future work.

Figure 19 presents examples from our HourHDVC dataset alongside predictions generated by our model. The scene captions effectively incorporate context through visual descriptions and speech. These results demonstrate that our method successfully performs context-aware scene dense video captioning. Additionally, we generate a video narrative that accounts for long-term dependencies across all scenes in hour-long videos.

# G LLM ASSISTANCE

Large Language Models (LLMs) were employed to support translation and light editing.

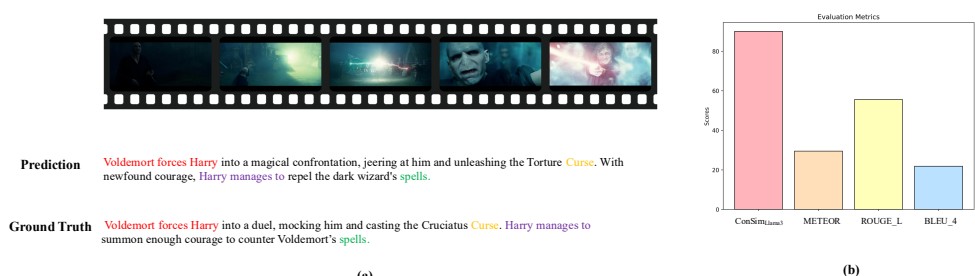

(a)
(b)

Figure 13: **Comparison of our ConSim$_{Llama3}$ with N-gram based metrics.** In (a), we compare two semantically identical sentences using both N-gram-based metrics and our ConSim$_{Llama3}$. In the two sentences, words highlighted in the same color represent identical words. We normalized all metrics to a uniform range of 0 to 100. In this comparison, ConSim$_{Llama3}$ measures a score of 90, while METEOR, Rouge-L, and BLEU4 scored 29.48, 55.49, and 21.85, respectively, in (b). These results show that our ConSim$_{Llama3}$ goes beyond simple word matching and can evaluate contextual semantic similarity.

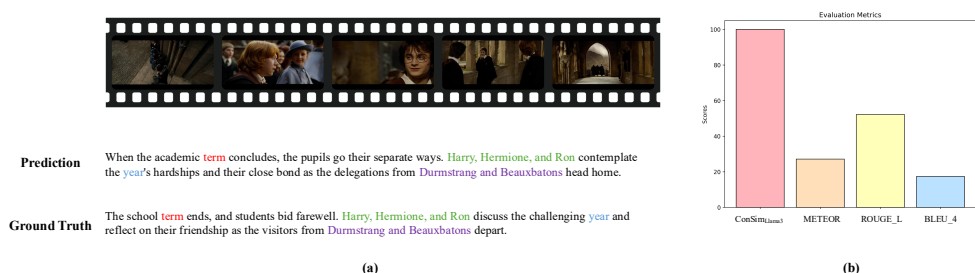

(a)
(b)

Figure 14: **Comparison of our ConSim$_{Llama3}$ with N-gram based metrics.** In (a), we compare two semantically identical sentences using both N-gram-based metrics and our ConSim$_{Llama3}$. In the two sentences, words highlighted in the same color represent identical words. We normalized all metrics to a uniform range of 0 to 100. In this comparison, ConSim$_{Llama3}$ measures a score of 100.00, while METEOR, Rouge-L, and BLEU4 scored 27.16, 52.22, and 17.38, respectively, in (b). These results show that our ConSim$_{Llama3}$ goes beyond simple word matching and can evaluate contextual semantic similarity.

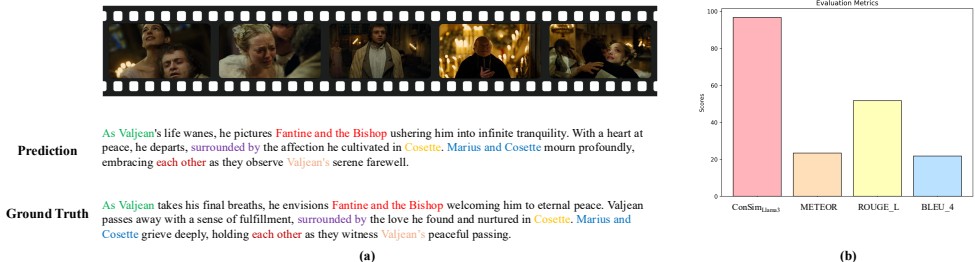

(a)
(b)

Figure 15: **Comparison of our ConSim$_{Llama3}$ with N-gram based metrics.** In (a), we compare two semantically identical sentences using both N-gram-based metrics and our ConSim$_{Llama3}$. In the two sentences, words highlighted in the same color represent identical words. We normalized all metrics to a uniform range of 0 to 100. In this comparison, ConSim$_{Llama3}$ measures a score of 96.66, while METEOR, Rouge-L, and BLEU4 scored 23.36, 51.69, and 21.78, respectively, in (b). These results show that our ConSim$_{Llama3}$ goes beyond simple word matching and can evaluate contextual semantic similarity.

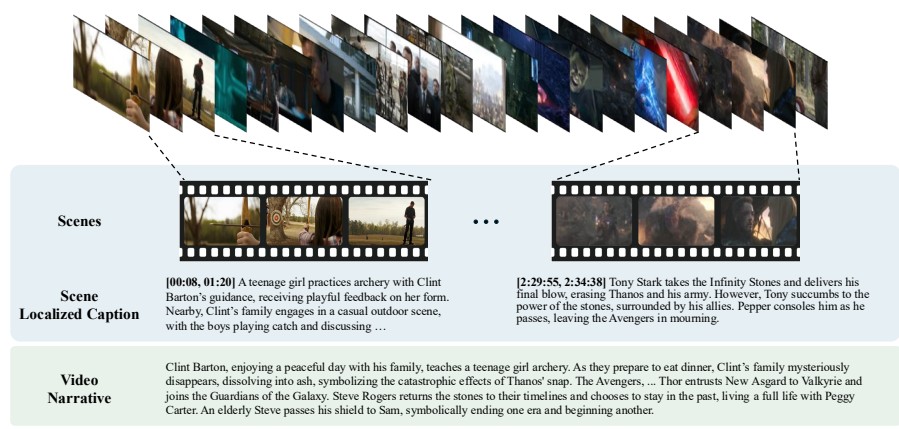

Figure 16: **Data Example in HourHDVC Train set.**

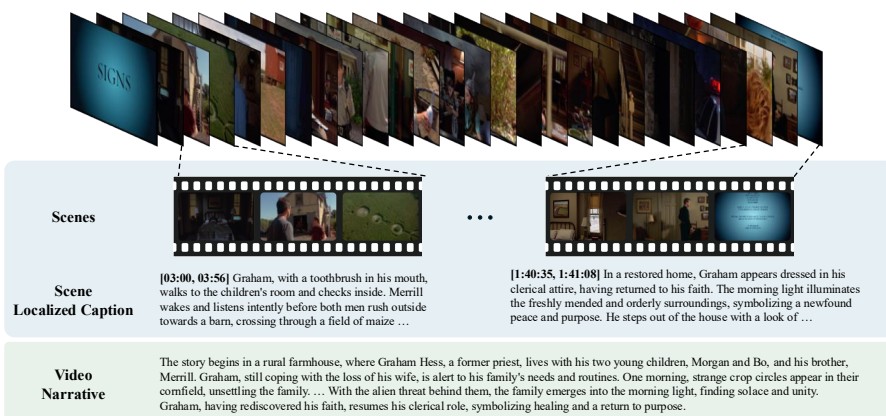

Figure 17: **Data Example in HourHDVC Eval set.**

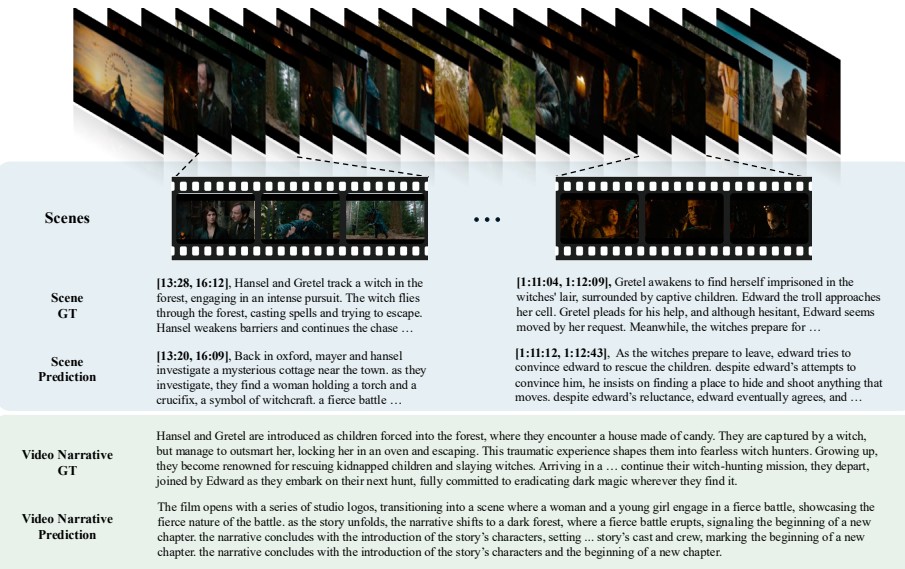

Figure 18: **Example of predictions on HourHDVC eval set.** For the scene dense video captioning, we evaluate the localization and captioning for the local scenes of an hour-long video. For the video narrative captioning, we evaluate the overall context generated by the scene predictions.

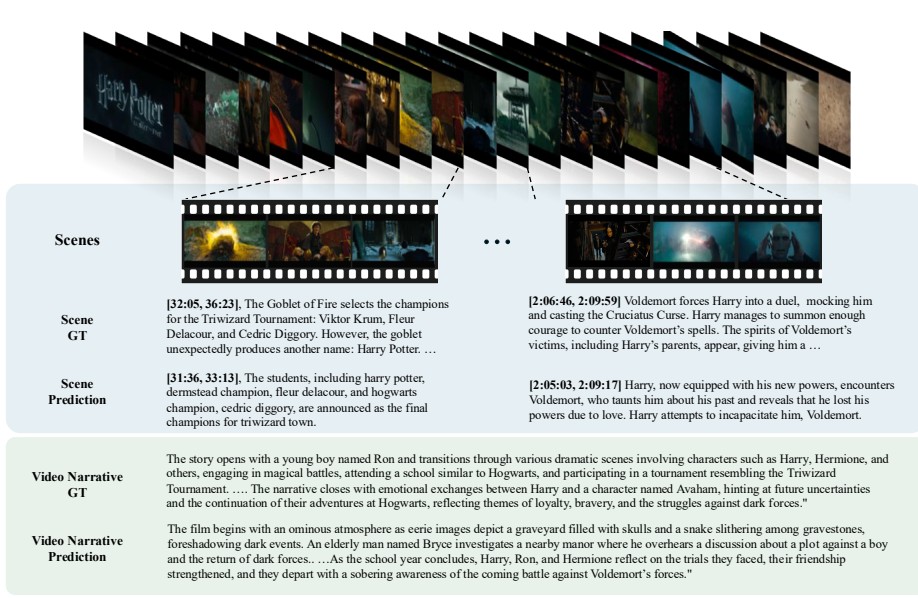

Figure 19: **Example of predictions on HourHDVC eval set.** For the scene dense video captioning , we evaluate the localization and captioniong for the local scenes of an hour-long video. For the video narrative captioning, we evaluate the overall context generated by the scene predictions.

**Algorithm 1** Python script for scene segmentation and description generation with our instruction to OpenAI GPT4o

```
[language=python,basicstyle=\scriptsize\ttfamily]
def process_events_in_batches(data,movie_name,client=ChatGPTClient(driver)):
    duration = data[movie_name]['duration']
    #set batch 200 for precise generation of LLM
    batch_events = data[movie_name]['batch_events']
    prompt = f"""
    You are an expert in scene separation and scene summarization. You have prepared hundreds
        of sentences that consist of audio descriptions (AD) and speech transcripts from
        movies. Audio descriptions provide explanations of visual elements in a narrative
        manner. Therefore, when the speech timestamps overlap with the appearance times of
        the visual scenes, the timestamps of the audio descriptions are assigned after the
        speech, which may differ from the actual timestamps of the movie. The speech
        information is generated automatically using an ASR model, so there is a possibility
         of mismatch compared to the real video. You must effectively utilize both the
        speech information as the only scene context and the audio description information
        as time-noised and semantic visual information to construct the scenes.
    Movie Name: {movie_name}
    Total Movie Duration: {duration}
    Below is instruction for your scene summarization.

    **1.How to summarize? 2 step - Group connected incidents and summarize with grouped
        incidents**:
    First step - Separate the scene with grouping incidents) Scenes should be separated by
        different important events. Here, an important event refers to an occurrence that
        involves a single coherent context (e.g., when a single action or interaction lasts
        for an extended period, when conversations are continuously based on a similar theme
        , or when scenes in the same category persist at an abstract level, even if the
        detailed actions differ, etc.).
    Second step - Summarize with the grouped incidents) Then you must summarized the scene
        within the grouped incidents. For example, grouped results are scene1:[[45,55]
        sentence1, [57,200]sentence2, [200,212]sentence3], scene 2:[[222,225]sentence5,
        [240,251]sentence5]. Then you summarize scene 1 with sentence1-3, summarize scene2
        with sentence 4 and sentence5. Don't use other grouped incidents when summarize a
        scene.
    Second step - Summarize with the grouped incidents) Then you must summarized the scene
        within the grouped incidents. For example, grouped results are scene1:[[45,55]
        sentence1, [57,200]sentence2, [200,212]sentence3], scene 2:[[222,225]sentence5,
        [240,251]sentence5]. Then you summarize scene 1 with sentence1-3, summarize scene2
        with sentence 4 and sentence5. Don't use other grouped incidents when summarize a
        scene.
    **2.Use audio descriptions (AD) as the main time reference**: When merging audio
        descriptions with speech, prioritize the timestamps of the AD to correct any timing
        mismatches from speech recognition (ASR). Structure scenes based on this timing to
        create a cohesive narrative.
    **3.Keep the summary objective**: Do not include opinions or assumptions. Summaries must
        be factual and focus on describing the events and actions.
    **4.Summarize all scenes concisely:** Avoid overly detailed descriptions and use as few
        semantically compressed sentences as possible to convey the key events. Keep the
        focus on the most essential elements needed to understand the scene.
    **5.Utilize speech**: When utilize speech, summarize or simply paraphrase the dialogue
        instead of quoting it exactly. Focus on the meaning or intent behind the words.
    **6.Strict Time Boundaries**: Summarize the scene only within the grouped incidents
        provided. Do not project or hint at events that happen beyond the current scene.
        When summarizing a scene, stick to what is shown up to the last second of the scene.
         Refrain from mentioning characters' emotions, intentions, or events that occur
        after the defined end of the scene.
    **7.Avoid redundancy**: If similar scenes are part of the same overarching event or theme
        , group them as one scene to avoid stemming the tide. Then we can get a continuous
        single scene. Treat minor variations in action or dialogue as part of the same scene
         if they share the same context or objective.

    Please summarize the scenes for the following events:
    {batch_events}

    Output format is below:
    Step1 - Grouping incidents to Scene :
    <Grouping Start>
    [Scene n]
    Grouping Theme1 : 1or2 words
    Time : [Start time,End time]
    Grouped Audio Description Index : [Start Index, End Index]
    <Grouping End>

    Step2 - Scene Summarization:
    [Start of Scene n]
    Timestamp: [start_time, end_time]
    Summary: ~
    [End of Scene n]
    You must strictly follow the 8 instructions provided for scene summarization without
        exception. Especially, think about it step-by-step that you follows the 1,3,4,6
        instruction, then answer me.
    """
    client.enter_prompt(prompt)
    response = client.get_latest_response()
```

1404
1405
1406
1407
1408
1409
1410
1411
1412
1413
1414
1415

**Algorithm 2** Python script for video narrative generation with our instruction to OpenAI GPT4o

```python
[language=python,basicstyle=\scriptsize\ttfamily]
def process_events_in_batches(data,movie_name,client=ChatGPTClient(driver)):
    duration = data[movie_name]['duration']
    events = data[movie_name]['events']

    prompt = f"""

    Movie Name: {self.movie_name}

    You are an expert in scene summarization. You have prepared tens of paragraphs that
        consist of scene descriptions from movies.

    I will provide you with sets of scene description paragraphs from hours-level videos.

    Below are instructions for your scene summarization.

    1. How to summarize? - Three-Act Phase Structure ( Setup-Confrontation-Resolution. Each
        Phase should be seperated with different paragraphs.
    Setup: Exposition , Introduction of conflict.
    Confrontation: Rising Action, Climax.
    Resolution: Falling Action, Denouement.

    2. Your Role: Summarize with the given paragraphs. You must summarize by focusing on the
        overall storyline and context of the video.

    3. Keep the summary objective: Do not include opinions or assumptions. Summaries should
        be factual and focus on describing the events and actions.

    4. Summarize concisely: Avoid overly detailed descriptions and use as few semantically
        compressed sentences as possible to convey the key events. Keep the focus on the
        most essential elements needed to understand the whole video.

    Please summarize the scenes for the following events:
    {events_text}

    Output format is below:

    [Start of Summarization]
    Setup:
    Confrontation:
    Resolution:
    [End of Summarization]
    Think about it step-by-step that you follows the 1,2,3,4 instruction, then answer me.
    """
    client.enter_prompt(prompt)
    response = client.get_latest_response()
```

1448
1449
1450
1451
1452
1453
1454
1455
1456
1457

**Algorithm 3** Python script for scene segmentation and description generation with naive instruction to OpenAI GPT4o

```python
[language=python,basicstyle=\scriptsize\ttfamily]
def process_events_in_batches(data,movie_name,client=ChatGPTClient(driver)):
    duration = data[movie_name]['duration']
    events = data[movie_name]['events']

    prompt = f"""

    Prepared hundreds of sentences are event captions of a movie video.
    I will give you sets of events that include timestamps and captions of events.
    Separate all the given sets of events into scenes, and make summaries and merged
        timestamps for each scene.
    Scenes should be separated by different places and cover all the events.
    Each summary can consist of several sentences or one sentence if necessary.

        Output format is below:
        Scene n
        Timestamp: [start_time, end_time]
        Summary: ~

    Movie Name: {movie_name}
    Total Movie Duration: {data[movie_name]['duration']}
    Critical Condition: Make the all of summarization result that covers all Movie duration.
    Making sure this condition is very important.
    {events}
    """
    client.enter_prompt(prompt)
    response = client.get_latest_response()
```

**Algorithm 4** Python script for ConSim, with Llama3-70B

```python
import transformers
import torch
import re

def evaluate_similarity_withLLM(prediction, ground_truth):

    #set LLM model
    pipeline = transformers.pipeline(
        "text-generation",
        model="meta-llama/Meta-Llama-3-70B-Instruct",
        model_kwargs={"torch_dtype": torch.bfloat16},
        device_map="auto",
    )

    # Prepare Prompt
    prompt = f"""
You are an intelligent chatbot designed for evaluating the quality of generative outputs
    for movie scene descriptions. Your task is to compare the predicted scene
    descriptions with the correct scene descriptions and determine its level of match,
    considering mainly the semantic similarity for the scene understanding.

Here's how you can accomplish the task:

**1. Semantic Similarity for Scene Understanding**: Evaluate whether the predicted scene
    description effectively captures the semantic information of the ground truth (GT)
    without relying on the specific words or expressions of the GT. The evaluation
    should consider whether the scene can be understood. Therefore, consider synonyms or
     paraphrases as valid matches.

**2. Person Matching**: Consider pronouns like 'he' or 'she' as valid matches with
    character names.

**3. Evaluation Method**: Evaluate each item with 1-10 points.
1) "Semantic Similarity": What is the level of semantic similarity between the scene
    description prediction and the ground truth (GT)?
2) "Temporal Alignment": Even if the prediction's scene and the GT's scene point to
    different times, is the description about a scene with the same theme?

This is an example of evaluation situation.
Given set:
Prediction: [126.568, 341.12], In a tranquil rural setting, a family photo is shown in a
    house. A man goes through his morning routine and visits the children's room.
    Noticing the absence of the children, the man, along with another man, runs towards
    the sound of the children in the cornfield. They find the daughter between the split
     corn rows...
Matched_gts: [180.0, 328.616], Graham, with a toothbrush in his mouth, walks to the
    children's room and checks inside. Merrill, a younger man, wakes suddenly and
    listens intently before both men rush outside towards a barn, crossing through a
    field of maize. They find Graham's daughter, Bo...

(Q) What is the final evaluation score for the given example set?
(Reasoning) To answer the question, we need to consider three aspects.
1) "Semantic Similarity": The prediction's description is semantically similar to the GT,
     even though the detailed descriptions and expressions are different. (Score: 9)
2) "Temporal Alignment": The prediction covers a time earlier than the GT, so it
    describes scenes not expressed in the GT. However, in the overlapping time interval,
     it describes the same themed scene. (Score: 8)
(A)
Considering each evaluation criterion, the result is ...
Evaluation Criteria: [9,8], Final Score:8.5 .

(Q) Then, what is the evaluation result for the given prediction and GT below?

Prediction:
{prediction}
GT:
{gt}

Your output form is like below:
Evaluation Criteria: [score1, score2], Final Score: average of Score1-Score2 .
(A)
"""

    # Llama evaluatioin
    outputs = pipeline(prompt)
```

**Algorithm 5** Python script for ConSim for narrative captioning, with Llama3-70B

```python
import transformers
import torch
import re

def evaluate_similarity_withLLM(prediction, ground_truth):
    #set LLM model
    pipeline = transformers.pipeline(
        "text-generation",
        model="meta-llama/Meta-Llama-3-70B-Instruct",
        model_kwargs={"torch_dtype": torch.bfloat16},
        device_map="auto",
    )
    # Prepare Prompt
    prompt = f"""
You are an intelligent chatbot designed for evaluating the quality of generative outputs
    for movie scene descriptions. Your task is to compare the predicted scene
    descriptions with the correct scene descriptions and determine its level of match,
    considering mainly the semantic similarity for the scene understanding.

Here's how you can accomplish the task:
**1. Semantic Similarity for Scene Understanding**: Evaluate whether the predicted scene
    description effectively captures the semantic information of the ground truth (GT)
    without relying on the specific words or expressions of the GT. The evaluation should
    consider whether the scene can be understood. Therefore, consider synonyms or
    paraphrases as valid matches.
**2. Person Matching**: Consider pronouns like 'he' or 'she' as valid matches with character
    names.
**3. Evaluation Method**: Evaluate each item with 1-10 points.
1) "Semantic Similarity": What is the level of semantic similarity between the scene
    description prediction and the ground truth (GT)?

These are examples of evaluation situation.
Given set 1 (Score = 3)
"prediction": The movie opens with a view of Earth from space, transitioning to a starry
    night sky with a view of Earth from space. The scene transitions to a beach, where a
    young boy, charlie, is seated on a beach while a young boy, sam, is on a beach. The
    scene transitions to...
"matched_gts": Charlie and Sam, two brothers, share a strong bond, highlighted by their love
    of sailing and shared dreams of adventure. During a sailing competition, they narrowly
    avoid obstacles and celebrate their victory together, reinforcing their commitment to
    each other. Their bond is further emphasized at an award ceremony and during...

(Q) What is the final evaluation score for the given example set?
(Reasoning) The prediction does mention Charlie and Sam, a beach, and a reference to an
    accident so some elements match. However, Sams death and the key  ghost  aspect
    are missing.

 Semantic  Similarity: Minimal alignment, only partial mention of characters/accident,
    lacking the core plot. (Score: 3)
(A)
Considering each evaluation criterion, the result is ...
Evaluation Criteria: [3], Final Score:3

Given set 2 (Score = 4)
"prediction": The story opens with a view of a cityscape surrounded by meteor showers
    descending from tokyo, followed by a report of an unidentified enemy reaching the
    worlds coastlines. A report explains that the world is at war, describing a
    coordinated attack on multiple locations across ...
"matched_gts": News reports reveal an alarming meteor shower off Tokyos coast, which is
    soon identified as a coordinated extraterrestrial attack. The global situation escalates
    as an unknown alien force invades twelve coastal regions, prompting urgent military
    mobilization. Marines at Camp Pendleton are briefed about the alien landings, observing
    images of mechanical alien structures...

(Q) What is the final evaluation score for the given example set?
(Reasoning) They align on the concept of alien invasion, Marines in LA responding to the
    threat, but many specific details like the named Staff Sergeant or the final command-
    center battle are omitted.

 Semantic  Similarity: Some main points are shared (alien landing, LA battle, Marines),
    but the finer details are missing. (Score: 5)
(A)
Considering each evaluation criterion, the result is ...
Evaluation Criteria: [4], Final Score:4
    """

    # Llama evaluatioin
    outputs = pipeline(prompt)
```

