# OpenReview forum: "How Do You Watch a Movie? HourHDVC: Hour-Long Hierarchical Dense Video Captioning"
_ICLR.cc/2026/Conference — Submitted to ICLR 2026_

### Official Review · Reviewer_jmSY · 2025-10-29

**Soundness:** 4
**Presentation:** 4
**Contribution:** 4
**Rating:** 10
**Confidence:** 3

**Summary:**

This paper introduces Hierarchical Dense Video Captioning (HDVC), a new task requiring both timestamped scene descriptions and an overall narrative for hour-long videos. To support HDVC, the authors present the HourHDVC dataset with 498 richly annotated movie videos, and propose LOCO—an end-to-end model that uses dual memory mechanisms to maintain coherence across long videos. LOCO significantly outperforms existing baselines on both scene and narrative captioning, as measured by the proposed ConSimLlama3 metric, which shows strong human correlation.

**Strengths:**

This work demonstrates significant strengths across multiple dimensions. It introduces a novel and well-defined task (HDVC) supported by the first large-scale, richly annotated dataset (HourHDVC) for hour-long videos. The proposed LOCO model is technically innovative, featuring an elegant two-tier memory design that effectively handles long-range dependencies, which is validated through extensive ablations. LOCO achieves substantial and consistent empirical improvements, establishing a new state-of-the-art on both scene and narrative captioning. The claims are further solidified by thorough evaluation, including strong baseline comparisons and a careful validation of both the dataset quality and the proposed ConSimLlama3 metric. Finally, the paper itself is exceptionally well-written and clearly structured, enhancing its overall impact and accessibility.

**Weaknesses:**

This work has several notable limitations. The HourHDVC dataset relies on GPT-4 for annotation, which may introduce LLM-specific biases or errors despite its high overall quality. The dataset and method are also primarily validated on movie content, leaving their generalizability to other long-form video genres less certain. Computationally, the complexity and cost of LOCO's multi-component architecture are not analyzed, raising scalability concerns. While the proposed ConSimLlama3 metric is well-validated, the near-zero BLEU scores and reliance on this single LLM-based metric suggest the evaluation scope could be broader. Finally, the main text lacks an explicit discussion of these limitations, which would provide a more balanced presentation.

**Questions:**

1. The dataset pipeline relies on movie transcripts and audio descriptions. How would the approach handle videos without transcripts or with noisy speech (e.g. background music, crowd noise)? Could the model still perform well if only visual features are available?

2. The paper uses a three-act structure for narrative generation. How robust is this scheme across different movie genres (e.g. non-fiction documentaries or experimental films)? Did the authors try alternative narrative segmentations?

3. What is the computational cost of training and inference with LOCO, compared to baseline models? Can it run in reasonable time on a single long video?

4. For scene segmentation and captioning, how sensitive is the performance to the quality of the initial transcript? (E.g. if ASR transcripts have errors, does LOCO degrade gracefully?)

5. Do the authors plan to release HourHDVC publicly? If so, will they provide the raw video clips, transcripts, or only the annotations?

6. Could the authors provide qualitative examples of scene-level captions and the generated narrative, especially highlighting cases where LOCO succeeds or still fails? This would help illustrate the model’s behavior.

7. Has LOCO been compared to using a large end-to-end multimodal model (e.g. GPT-4V) directly on the long video, perhaps with sliding windows? It would be interesting to see how a powerful MLLM stacks up with external memory.

8. In the controlled experiment on noise (Table 6), the precision is relatively low (~52%). Can the authors clarify whether missing or extra scene segments affect narrative quality?

---

> ### Author Response · Authors · 2025-11-21
>
> We appreciate your recognition of the novelty of the HDVC task, the technical innovation of LOCO’s two-tier memory, and the comprehensive evaluation of our dataset and metric. We have addressed your constructive feedback below to further strengthen the paper.
>
> **Potential LLM Biases and Errors in Dataset**
>
> Quantitative and qualitative analyses confirm that noise in LLM-generated data is minimal. To address concerns about LLM artifacts, we compared the pre-refinement (GPT-generated) and post-refinement (Human-refined) evaluation sets. For 364 scenes, the IoU of timestamps before and after refinement is 0.80, indicating high-quality localization despite minor misalignments. Furthermore, textual refinements were required for only 33 out of 364 scenes (approx. 9%), and the remarkably low hallucination rate (~1.9%) confirms that our pipeline generates high-quality descriptions even without human refinement. As shown in Table 6, the model trained on noisy data (Recall 29.58) showed comparable performance to the one trained on refined data (Recall 32.54), suggesting that while refinement yields minor improvements, the noise itself does not compromise the validity of the benchmark. Additionally, in our blind user study (Section 6.1), there was no statistically significant difference between GPT-generated captions (8.49) and expert-refined captions (8.51), verifying the scalability of our high-quality generation pipeline.
> Also, to rigorously address the concern of single-model bias, we conducted a cross-model validation using Gemini 2.5 Pro and GPT-5. For scenes with high temporal alignment (IoU
>  0.7), we compared their outputs against our GPT-4o results that used in our data generation pipeline. We compared their generated captions against our GPT-4o results for scenes with high temporal alignment (IoU > 0.7). The results showed low lexical overlap (Average BLEU-4: 7.90 vs. Gemini, 5.20 vs. GPT-5) but high semantic agreement (ConSim: 89.80 vs. Gemini, 87.40 vs. GPT-5.1). This combination of diverse wording (low BLEU) and identical meaning (high ConSim) confirms that our pipeline extracts objective narrative facts grounded in the source inputs (Transcript/AD), rather than reflecting the stylistic bias of a single model.
>
> **Computational Complexity and Scalability of LOCO**
>
> LOCO is designed for efficiency on consumer-grade hardware. We pre-extract CLIP ViT-L/14 features and process sliding windows independently. LOCO can be trained on a single 24GB GPU (e.g., RTX 4090), and inference for an hour-long video takes only ~36 seconds; for comparison, the end-to-end backbone without the memory module takes ~32 seconds, and a simple variant without the sliding-window system runs in ~5 seconds. With only one learnable memory token per level, the memory overhead is minimal, confirming LOCO’s practicality and scalability for long-form video understanding.
>
> **Evaluation Scope and Discussion on N-gram Limitations**
>
> We appreciate the reviewer's insight regarding the evaluation scope. We clarify that our evaluation framework is already designed to be comprehensive, incorporating not only ConSim but also traditional metrics like METEOR, and F1 (for localization) as reported in the paper. The near-zero BLEU scores are an expected phenomenon due to the "one-to-many" nature of long-form narrative generation, where semantically identical stories can be expressed with vastly different vocabularies, rendering rigid N-gram matching ineffective. Also in Supplementary Material Section E, we demonstrates that traditional metrics (e.g., METEOR) are overly sensitive to noise (dropping to ~52 points with 10% perturbation) while ConSim remains robust.
>
> **Robustness to Noisy Speech**
>
> To test robustness against noisy speech and varying degrees of audio dependence, we analyzed two contrasting movies from our evaluation set: Les Misérables, a musical with frequent songs and heavy background music where 46.7% of speech segments overlap with background music, and Signs, where background music overlaps with speech for only 18.5% of the speech duration, resulting in comparatively clean and distinct speech whenever it occurs. We evaluated the model’s inference performance (averaged over four tIoU thresholds: 0.1, 0.3, 0.5, and 0.7) on these two movies. Signs achieved METEOR 0.0560, CIDEr 0.0306, ROUGE-L 0.0713, and BLEU-4 0.0065. In comparison, Les Misérables recorded METEOR 0.0551, CIDEr 0.0245, ROUGE-L 0.0574, and BLEU-4 0.0068. As expected, we observed a slight performance decrement in metrics such as CIDEr and METEOR in the highly noisy environment of Les Misérables. However, the model maintains competitive captioning capability without catastrophic failure. This trend suggests that while clear speech transcripts contribute to optimal performance, LOCO effectively leverages visual cues to mitigate the impact of compromised audio modalities or sparse dialogue

---

> ### Author Response · Authors · 2025-11-21
>
> **Sensitivity to ASR Transcript Quality**
>
> We analyzed the sensitivity of our model to transcript completeness by simulating varying levels of ASR quality through a random transcript drop mechanism. As shown in Table, the model exhibits a degree of resilience against partial information loss. Interestingly, performance remains relatively stable or even slightly improves up to a 0.3 drop rate (CIDEr 3.32, METEOR 6.15) compared to the 'No drop' baseline (CIDEr 3.20, METEOR 6.41). However, performance begins to decline notably as the drop rate exceeds 0.5, eventually reaching CIDEr 1.41 and METEOR 4.03 at a 0.9 drop rate. This trend indicates that while high-quality transcripts are essential for peak performance, the model can withstand a significant amount of noise (up to ~30% loss) without degradation, though severe sparsity (90% loss) does impact the detailed captioning quality.
> | Subtitle Drop Rate | CIDEr | METEOR | ROUGE-L | F1    |
> |--------------------|-------|--------|---------|-------|
> | No drop            | 3.20  | 6.41   | 8.00    | 45.09 |
> | 0.1                | 2.96  | 6.29   | 7.62    | 42.61 |
> | 0.3                | 3.32  | 6.15   | 7.91    | 45.34 |
> | 0.5                | 2.53  | 5.69   | 7.33    | 41.33 |
> | 0.7                | 1.67  | 5.01   | 6.78    | 39.66 |
> | 0.9                | 1.41  | 4.03   | 5.63    | 36.59 |
>
>
> **Data Release**
>
> Since our dataset is built upon MAD, we respect the original licensing. We will publicly release all annotations (timestamps, captions, narratives), and scripts to align our data with the MAD video files. This ensures the benchmark is reproducible while complying with copyright regulations.
>
> **Success vs. Failure Examples**
>
> Supplementary Figure 19 (Harry Potter) illustrates LOCO’s success in integrating visual context to capture complex plots and specific characters. ("The Goblet of Fire selects... unexpectedly produces another name: Harry Potter"), showing the integration of visual events with context. Conversely, Figure 18 shows a failure case where the model suffers from repetition loops ("the narrative concludes with...the narrative concludes with..."), highlighting the difficulty of the HDVC task. This suggests that while LOCO sets a strong baseline, the benchmark remains challenging for future research.
>
> **Comparison with MLLMs**
>
> We did not use large commercial MLLMs (e.g., GPT-4V) for the main comparison due to two key issues: data leakage and task suitability. As detailed in Supplementary Section C (Figure 9), models like GPT-4o already possess pre-existing knowledge of the movies in our dataset (e.g., being able to describe the plot of Signs without any video input), which undermines any fair, vision-based evaluation. Moreover, these models currently cannot realistically ingest untrimmed hour-long videos to autonomously perform precise dense localization and captioning without aggressive chunking and heuristic post-processing. Instead, we evaluated accessible open-source Video-LLMs (~7B parameters) such as VTimeLLM and TimeChat (Table 2), but they performed poorly, failing to maintain long-term context over the full movie. We also tested the “clip-wise captioning and merging via sliding window” approach; however, it proved inferior to LOCO because simple merging fails to capture the temporal dependencies and global narrative coherence essential for hour-long video understanding. Note that, while some degree of pre-training exposure to similar content cannot be entirely ruled out for open-source 7B-scale Video-LLMs, this does not weaken our conclusions. We include these models not as leakage-free oracles, but as the most relevant and reproducible baselines for dense video captioning at a realistic scale. If anything, any residual prior knowledge they may have about specific movies works in their favor, not ours. Yet they still perform markedly worse than LOCO on HourHDVC, suggesting that the main bottleneck in hour-long video captioning is not a lack of world knowledge, but the difficulty of maintaining long-range temporal structure and performing precise scene localization and summarization over very long contexts.

---

> ### Author Response · Authors · 2025-11-21
>
> **On Scene Precision, Training Noise, and Narrative Quality.**
>
>  Table 6 is designed as a controlled experiment to quantify how timestamp noise in the training annotations affects scene dense video captioning. Although the scene localization precision is relatively low (around 52%), the overall scene-level captioning performance remains stable. The model trained on noisy, LLM-generated timestamps achieves CIDEr 1.15, METEOR 4.90, and ROUGE-L 6.16, which are not far from the scores of the model trained on human-refined timestamps (CIDEr 1.69, METEOR 4.84, ROUGE-L 6.78) for Scene Dense Video Captioning. This indicates that, even though a certain amount of timestamp noise is present in the training data, the model can still learn a reasonable correspondence between visual–linguistic context and scene descriptions, and such noise does not catastrophically degrade scene-level captioning performance.
>
> Regarding its impact on global narrative quality, we emphasize that the narrative decoder does not rely on a single “correct” set of scene boundaries; instead, it consumes all predicted scene representations as memory. In other words, even when the scene precision is not perfect, the narrative module has access to dense predictions over the entire timeline and can integrate information from multiple overlapping or slightly misaligned segments. This design inevitably allows some error propagation from the scene stage, but at the same time makes the narrative summarization relatively robust. The memory mechanism aggregates redundant evidence and focuses on recurrent, salient events and the overall flow of the story, rather than depending on any single boundary decision. Moreover, because all scene-level predictions are retained as memory, there are very few regions on the timeline where information is completely missing, which helps preserve the contextual flow at the narrative stage. Consequently, while timestamp noise can introduce local errors at the scene level, our design of using dense scene predictions as a rich memory largely mitigates the impact of this relatively low precision on the global narrative quality and coherence.
>
>
> **Three-Act Structure of Narrative Generation**
>
> Our global narrative summary follows the classical concept that a story has a beginning, middle, and end[1], instantiated as the Three-Act Structure (Setup–Confrontation–Resolution)[2,3]. This structure is fundamental to storytelling, making it robust across genres, including documentaries and experimental films, which almost invariably possess a narrative arc. While other formats like the 5-act structure exist, the 3-act structure is a universal foundation that generalizes well across various long-form video contents.
>
> [1] Aristotle, Translation. Poetics. Vol. 9. Oxford: Clarendon Press, 1968.
>
> [2] Field, Syd. Screenplay: The foundations of screenwriting. Delta, 2005.
>
> [3]Trottier, David. The screenwriter's bible: A complete guide to writing, formatting, and selling your script. Vol. 5. Los Angeles: Silman-James Press, 1998.

---

### Official Review · Reviewer_1GSZ · 2025-10-30

**Soundness:** 2
**Presentation:** 2
**Contribution:** 3
**Rating:** 4
**Confidence:** 3

**Summary:**

Previous work struggles with hour-long videos because existing datasets fail to capture long-range context, and current models are not designed to handle such extensive temporal dependencies.
To address this limitation, the authors introduce a new task called Hierarchical Dense Video Captioning (HDVC) for long-form video understanding. HDVC requires generating both scene-level captions and a global narrative caption for the entire video. They also release HourHDVC, a new dataset that supports this task.
In addition, the authors propose LOCO (LOng COntext memory-based hierarchical dense video captioning), a model that uses a two-tier memory system, Context-aware Memory and Long-term Context Memory, to maintain coherent narratives over extended time spans.

**Strengths:**

1. The authors introduce a new task and dataset with multi-level annotations, providing scene-level and video-level narrative captions. This dataset is likely to be valuable for future research on long-form video understanding.
2. They present a new model, LOCO, which outperforms existing methods on the HourHDVC dataset.
3. They also propose a new evaluation metric, ConSim, designed to measure how well models capture contextual information.

**Weaknesses:**

1. The paper lacks sufficient detail regarding the LOCO architecture. In particular, it is unclear which backbone models are used for each component (e.g., the temporal encoder), whether the framework generalizes to alternative model choices, and what hyperparameters are used during training. More implementation details would help clarify the method and improve reproducibility.
2. It is also unclear why the performance of Scene Dense Video Captioning decreases when context-aware memory is added in Table 4. Since this component is intended to enhance the modeling of long-range context, additional explanation is needed to understand why it does not consistently improve performance.
3. It seems like the comparison setup is unfair for Table 2: the proposed method is trained in-domain, whereas some baselines may not be.
4. More detail is needed about the human refinement of the evaluation set. For example, it is unclear what specific changes annotators made, how frequently edits were applied, who the annotators were, and what procedures were used to ensure annotation quality.
5. The description of the ConSim metric is also incomplete. How ConSim is computed? Is it a form of LLM-as-a-judge–style evaluation?
6. The paper does not report results from other methods on the Ego4D-HCap dataset (Table 5). Could you also add the results of different methods?

**Questions:**

1. It appears that the context-aware memory is not updated across scenes, and instead each scene has its own separate context-aware memory. Is this correct? If so, if the video gets long, wouldn't it need large storage?
2. In line 290, when you mention parameter sharing, does this mean that the same LoRA weights are shared, or that it is the same base model with different LoRA adapters for each component? The figure only shows LoRA applied to the video-narrative captioning module, so clarification would be helpful.
3. In Equation (2), I assume M represents memory, but it would improve readability if all symbols were explicitly defined in the text.

---

> ### Author Response · Authors · 2025-11-21
>
> We thank the reviewer for the positive feedback and constructive comments. We will include clearer notations (e.g. M).
>
> **Implementation Details & Memory Storage**
>
> LOCO uses a T5-Base backbone with a non-overlapping window size of 600 (1fps). And our baseline is built on the Vid2Seq architecture trained on VidChapters, with the model containing 315M trainable parameters. All experiments were conducted on four RTX 4090 GPUs. To verify Memory Storage efficiency, we correct a misunderstanding: Context-aware Memory is a single shared learnable token across scenes, not per-scene. Thus, storage overhead is negligible. (See Supp. A for full setup).
>
> **Impact of Context-aware Memory on Scene DVC**
>
> The performance drop observed in Table 4 is attributed to the trade-off inherent in the End-to-End multi-task learning setup, where the model aims to optimize both Video Narrative and Scene DVC simultaneously without Long-Term Memory. To verify the effectiveness of the Context-aware Memory in isolation, we conducted an experiment training only for Scene Dense Video Captioning, excluding the Video Narrative task. The model trained with Context-aware Memory achieved METEOR 6.51 and F1 45.08, showing a clear improvement over the baseline without memory (METEOR 6.32, F1 43.96). This confirms that Context-aware Memory effectively enhances scene-level understanding when not constrained by the multi-task optimization trade-off.
>
> **Fairness of Comparison in Table 2**
>
> We clarify that TimeChat and VTimeLLM were evaluated in a Zero-shot setting without fine-tuning on HourHDVC. This decision was made because these models utilize different data input formats(e.g. multi-turn chat), and re-generating the entire dataset to fine-tune them would create a different form of unfairness; furthermore, TimeChat is originally designed to report zero-shot performance. To ensure transparency and a fair comparison, we will modify Table 2 to explicitly label these baselines as "Zero-shot", clearly distinguishing them from the fine-tuned methods.
>
> **Details on Human Refinement Process**
>
> Our refinement process involved two stages of expert verification to ensure high quality: a first field expert corrected timestamps and captions (Stage 0 to 1), and a second expert performed a more detailed refinement (Stage 1 to 2). To quantify the extent of the refinement, we measured the Text Refinement Ratio (proportion of events where text was modified) and the Timestamp Intersection-over-Union (IoU) between the pre- and post-refinement intervals. In the first stage, the Text Refinement Ratio was 0.07 (26/364 events) with a Timestamp IoU of 0.80 (comparing Stage 0 vs. Stage 1). In the second stage, the Text Refinement Ratio was 0.02 (7/364 events) with a Timestamp IoU of 0.99 (comparing Stage 1 vs. Stage 2). Overall, a total of 33 events were modified (Total Refinement Ratio: 0.09). The changes were categorized into omission of specific details (9 cases), hallucination (7 cases), and overlapping descriptions between scenes (17 cases); the remarkably low hallucination rate (~1.9%) confirms that our pipeline generates high-quality natural language descriptions even before human refinement.
>
> **Details on ConSim Metric**
>
> Yes, ConSim is an LLM-as-a-judge style evaluation, and as detailed in Supplementary Material Section E, we ensure reliability through specific evaluation criteria (such as semantic information matching) and by inducing step-by-step reasoning (Chain of Thought). We repeat this evaluation three times and average the scores to achieve a stable metric, noting that temporal alignment is assessed during Scene DVC but not evaluated in the Video Narrative Captioning stage. The ConSim showed the highest correlation with human judgment (METEOR: 0.27, ROUGE_L: 0.07, ConSim_Llama3: 0.87, ConSim_Llama3.3: 0.75, ConSim_Gpt5.1: 0.84)
>
> **Comparison on Ego4D-HCap**
>
> We clarify that existing methods reported on Ego4D-HCap are not designed for the Dense Video Captioning (DVC) setting, preventing direct use of their reported numbers. Instead, we trained previous methods on the Ego4D-HCap dataset (filtered for videos > 1 hour) under our DVC setting for a fair comparison. LOCO demonstrates superior performance (CIDEr 206.49, METEOR 23.29) compared to PDVC(ICCV2021') (CIDEr 162.08, METEOR 19.93) and HiCM2(AAAI2025')(CIDEr 113.88, METEOR 16.69), confirming that LOCO maintains competitive performance and generalization capabilities across different domains.
>
> **Clarification of LoRA use**
>
> We clarify that LoRA is applied only to the decoder to enable a shared encoder-decoder architecture capable of interpreting global narratives while maintaining local scene understanding. Crucially, during the Scene stage, the LoRA module is not forwarded (and thus not updated), ensuring it does not affect scene predictions; it is activated and updated only during the Global Narrative stage to improve prediction and represent the narrative.

---

> > ### Comment · Reviewer_1GSZ · 2025-11-27
> >
> > Thank you for your detailed response. I have a few remaining questions:
> >
> > W1. Have you experimented with different backbone models, since T5 encoder is relatively old? Additionally, how does the choice of window size ($k$) in Equation 5 affect model performance?
> > W2. If there is an inherent trade-off in the end-to-end multi-task learning setup that leads to lower performance when the Context-aware Memory is added for Video Narrative, I am unclear about the necessity of including this component. Is its main purpose simply to construct inputs for the long-term memory module?
> > W6. Have you compared the compute cost of Vid2Seq and LOCO

---

> > > ### Author Response · Authors · 2025-12-02
> > >
> > > We thank the reviewer for these thoughtful follow-up questions and address each point in detail below.
> > >
> > >
> > > **Backbone Choice**
> > >
> > > LOCO follows the same initialization as the Vid2Seq baseline, a widely adopted SOTA model in recent Dense Video Captioning research. Vid2Seq uses T5 encoder pretrained on the C4 dataset and then performs additional domain-specific pretraining on large-scale instructional video corpora (HowTo100M → VidChapters-7M), acquiring video–language alignment and timestamp-aware description capabilities that cannot be obtained from text-only T5 checkpoints. Because LOCO is designed to effectively leverage this domain-specialized pretrained knowledge and timestamp-aware description capabilities, we retain the same T5-Base backbone used in Vid2Seq. Moreover, Table 2 shows that Vid2Seq, Vid2Seq-SW, and LOCO (OURS) are all evaluated under the same pretrained initialization, enabling a clear comparison of structural contributions without the confounding effects of backbone changes. Importantly, LOCO’s performance gains do not stem from using a newer or larger backbone, but from our proposed hierarchical memory design. For example, in Scene Dense Video Captioning, LOCO shows consistent improvements over the window-based Vid2Seq-SW baseline in ConSim_Llama3 (18.49 → 33.04), METEOR (2.87 → 6.41), and F1 (40.16 → 45.09).
> > >
> > >
> > > **Window Size Analysis**
> > >
> > > We explored four window sizes in 300-second increments (300/600/900/1200s). The results are shown in the table below. We chose 600 seconds because it provides the most balanced performance for both Scene DVC (F1 45.09, METEOR 6.41) and Video Narrative Captioning (METEOR 9.59). The average scene duration is 131 seconds, so a 300s window may cause more scene boundary fragmentation, which corresponds with the lower F1 (42.68). Conversely, very large windows (900–1200s) lead to sparse frame sequences (≈9–12 seconds per frame), which can miss visually short events, aligning with the drop in Scene DVC F1. In addition to F1, the Scene DVC METEOR also remains competitive at 600s (6.41), while avoiding the degradation seen at 900–1200s. Overall, 600 seconds offers an empirically stable trade-off between preserving local event detail and capturing longer contextual structure.
> > >
> > > | Window Size (seconds) | Scene METEOR | Scene ROUGE | Scene F1 | Narrative METEOR | Narrative BLEU4 |
> > > |-----------------------|-------------|-------------|----------|------------------|-----------------|
> > > | 300                   | 6.72        | 7.86        | 42.68    | 9.07             | 0.17            |
> > > | 600(Ours)               | 6.41   | 8.00   | 45.09| 9.59         | 0.38        |
> > > | 900                   | 6.28        | 6.70        | 30.22    | 8.37             | 0.32            |
> > > | 1200                  | 6.17        | 8.38        | 29.95    | 9.12             | 0.55            |

---

> > > > ### Author Response · Authors · 2025-12-02
> > > >
> > > > **Complementary Roles of the Two Memory Modules**
> > > >
> > > > The two memory modules play complementary roles rather than serving as a simple input construction pipeline for the long-term memory. As shown in our earlier official comment (Impact of Context-aware Memory on Scene DVC), the context-aware memory provides direct benefits for Scene Dense Video Captioning, which becomes more evident when the model is trained only on the scene objective (METEOR 6.51, F1 45.08 vs. 6.32 and 43.96 without memory). This confirms that the context-aware memory contributes meaningful local cues, but its effect is partially reduced in the end-to-end setting due to the joint optimization with the more difficult narrative objective.
> > > > We also evaluated a model that uses only the long-term memory in the end-to-end setting. This variant improves Video Narrative Captioning (METEOR 9.23 vs. 7.70 for no memory), but its Scene DVC performance (METEOR 6.29, F1 42.82) remains lower than that of the proposed LOCO model (METEOR 6.41, F1 45.09), reflecting that the long-term memory primarily supports narrative-level reasoning rather than fine-grained scene modeling. When both memories are used together, the model benefits from both types of information: the context-aware memory improves local event representation, and the long-term memory captures global dependencies. As a result, the full model achieves the best overall performance across tasks (Scene DVC METEOR 6.41, F1 45.09; Narrative METEOR 9.59). Thus, the context-aware memory is not included merely to build inputs for the long-term memory but serves as an essential component that complements it by strengthening scene-level modeling.
> > > >
> > > > **Compute Cost Comparison with Vid2Seq**
> > > >
> > > > We compared the compute cost of Vid2Seq and LOCO under the same sliding-window setup, using pre-extracted CLIP ViT-L/14 features for both models on HourHDVC. With identical windowing, the inference time for a two-hour video increases only slightly from Vid2Seq-SW (32 seconds) to LOCO (36 seconds). For reference, a variant without the sliding-window mechanism runs in 5 seconds, but this speed comes with substantially lower accuracy (Vid2Seq: ConSim 13.46, METEOR 1.76, F1 2.56 vs. LOCO: ConSim 33.04, METEOR 6.41, F1 45.09). Training remains feasible on a single 24GB GPU (e.g., RTX 4090), and the additional memory components add negligible overhead, as LOCO uses only one learnable memory token per level. Overall, LOCO provides substantially better accuracy than the sliding-window Vid2Seq baseline (Vid2Seq-SW: ConSim 18.49, METEOR 2.87, F1 40.16 vs. LOCO: ConSim 33.04, METEOR 6.41, F1 45.09) while maintaining almost the same computational cost, indicating that LOCO is a practical and efficient compared to Vid2Seq-SW.

---

### Official Review · Reviewer_AHWX · 2025-10-31

**Soundness:** 2
**Presentation:** 2
**Contribution:** 2
**Rating:** 2
**Confidence:** 3

**Summary:**

This paper introduces HourHDVC, a benchmark for hierarchical dense video captioning of hour-long videos, and proposes the LOCO model, which leverages a two-tier memory system to generate both scene-level and narrative-level captions. The dataset is constructed using LLM-generated captions, with human refinement for evaluation splits. Experiments show that LOCO achieves strong results on this benchmark.

**Strengths:**

1. System integration: The benchmark and model are well-integrated and address a technical gap in long-form video captioning.
2. Experimental thoroughness: The experiments are comprehensive for the proposed benchmark, including ablations and human evaluations.

**Weaknesses:**

1. Motivation unclear: Given that LLMs can already generate high-quality captions for long videos (as shown by the authors’ own quality control), the necessity of this benchmark and the proposed modeling innovations is questionable. The paper does not convincingly show that existing methods (e.g., clip-wise captioning and merging) are insufficient.
2. Algorithmic novelty: The modeling techniques are incremental adaptations of existing ideas, not fundamentally new algorithms.
3. Synthetic data risks: Heavy reliance on LLM-generated captions may introduce artifacts or biases, which are not deeply analyzed.
4. Generalization and practical impact: The work is evaluated only on the proposed benchmark, with limited exploration of broader applicability or real-world utility.

**Questions:**

1. Necessity of the task: Can the authors provide evidence or analysis showing that simple baselines (e.g., sampling short clips, captioning with LLMs, and merging) are insufficient for hour-long video captioning? What unique challenges does HourHDVC address that cannot be solved by existing methods?
2. Real-world motivation: Are there real-world applications or user studies that demonstrate a need for hierarchical dense captioning of hour-long videos?
3. Comparison to simple pipelines: How does the proposed approach compare to a pipeline that uses LLMs to caption short segments and then merges or summarizes them?
4. Synthetic data quality: Are there systematic errors or artifacts in LLM-generated captions that affect model training or evaluation?
5. Generalization: Can the model and dataset be applied to other domains or tasks, or is the contribution limited to the proposed benchmark?

---

> ### Author Response · Authors · 2025-11-21
>
> We thank the reviewer for recognizing the value of our benchmark and model.
>
> **Clarification of task, HDVC**
>
> We believe the reviewer’s concern may stem from a conflation between our data construction pipeline and the HDVC task definition. To clarify, GPT-4o contributes only to text-based data generation and is never involved in video-based inference.
>
> 1. Task Distinction
>
> Data Construction (Text-to-Text): The LLM receives text-only inputs (timestamps, transcripts, and audio descriptions) and generates refined captions. This leverages the LLM's strong text summarization capability.
>
> HDVC Task (Video-to-Caption): The model receives an untrimmed hour-long video (avg. 7,200s) and audio as input. It must autonomously predict (i) temporal boundaries (localization) and (ii) scene/narrative descriptions without any oracle timestamps.
>
> 2. Why Existing Models Fail Current LLMs/MLLMs fail at HDVC due to three constraints:
>
> (A) Localization: Text-LLMs cannot process frames; Video-LLMs suffer from timestamp hallucinations in long streams.
>
> (B) Context Limits: Processing 2-hour videos exceeds standard context windows and computational budgets.
>
> (C) Narrative Inconsistency: As noted, simple "clip-wise merging" fails to capture long-term dependencies (e.g., an event at min 5 affecting min 100), resulting in disjointed summaries.
>
> 3. Empirical Verification: As shown in Table 2, standard Video-LLMs (e.g., VTimeLLM) and sliding-window baselines failed significantly (METEOR ~1.28, F1 <30.39) compared to LOCO (METEOR 6.41, F1 45.09). This confirms that our hierarchical memory is essential for capturing narrative structure, which current pipelines lack.
>
> Thus, HourHDVC addresses a unique challenge—narrative structure reasoning in long-form video—that cannot be solved by existing LLM-based pipelines.
>
>
> **Algorithmic Novelty**
>
> LOCO is a novel end-to-end architecture that jointly performs localization and summarization, unlike multi-stage priors (e.g., VideoRecap). Our core contribution is integrating a Two-tier Memory System (managing intra- and inter-scene contexts) with a Gradient Flow Mechanism. This design allows gradients to propagate across sliding windows, enabling the model to learn long-term dependencies that standard disconnected window approaches miss. While some components have been explored individually, integrating them to optimize for hour-long videos is a necessary and novel contribution.
>
> **Verification of Synthetic Data Quality**
>
> Quantitative and qualitative analyses confirm that noise in LLM-generated data is minimal and does not adversely affect model training. We analyzed artifacts by comparing the pre-refinement (GPT-generated) and post-refinement (Human-refined) evaluation sets.
> For 364 scenes, the IoU of timestamps before and after refinement is 0.80. This indicates sufficiently high-quality localization despite minor misalignments between speech transcripts and audio descriptions. Furthermore, textual refinements were required for only 33 out of 364 scenes (approx. 9%). The reasons were categorized into: omission of specific details (9 cases), hallucination (7 cases), and overlapping descriptions between scenes (17 cases). The remarkably low hallucination rate (~1.9%) confirms that our pipeline generates high-quality natural language descriptions even without human refinement.
>
> To rigorously address the concern of single-model bias, we conducted a cross-model validation using Gemini 2.5 Pro and GPT-5. For scenes with high temporal alignment (IoU $>$ 0.7), we compared their outputs against our GPT-4o results that used in our data generation pipeline. We compared their generated captions against our GPT-4o results for scenes with high temporal alignment (IoU > 0.7). The results showed low lexical overlap (Average BLEU-4: 7.90 vs. Gemini, 5.20 vs. GPT-5) but high semantic agreement (ConSim: 89.80 vs. Gemini, 87.40 vs. GPT-5.1). This combination of diverse wording (low BLEU) and identical meaning (high ConSim) confirms that our pipeline extracts objective narrative facts grounded in the source inputs (Transcript/AD), rather than reflecting the stylistic bias of a single model.
>
> In the table 6, to analyze the effect of timestamp noise, we trained models separately on noisy (LLM-generated) and refined annotations for the same videos. The model trained on noisy data (Recall 29.58) showed comparable performance to the one trained on refined data (Recall 32.54). This suggests that while refinement yields minor improvements, the noise itself does not compromise the validity of the benchmark. And also in the blind test (Section 6.1), there was no statistically significant difference between GPT-generated captions (8.49) and expert-refined captions (8.51). This demonstrates that high-quality data can be generated without human refinement, ensuring the scalability of the dataset. Note that the evaluation set is a clean set that underwent two rounds of expert refinement.

---

> ### Author Response · Authors · 2025-11-21
>
> **Generalization to Other Domains (Ego4D-HCap)**
>
> We clarify that **the generalization experiment on Ego4D-HCap was already included in the main paper of the original submission**, and we provide additional explanation here for completeness. LOCO’s core architecture generalizes beyond movies to various long-form video domains. In Section 6.2, we validated LOCO's generalizability using the Ego4D-HCap dataset (containing videos >1 hour).  We clarify that existing methods reported on Ego4D-HCap are not designed for the Dense Video Captioning (DVC) setting, preventing direct use of their reported numbers. Instead, we trained previous methods on the Ego4D-HCap dataset (filtered for videos > 1 hour) under our DVC setting for a fair comparison. As shown in Table 5, LOCO(CIDEr 206.49, METEOR 23.29) achieved superior performance in the segment-level captioning task compared to Vid2Seq CIDEr 145.29, METEOR 19.49). We also compares LOCO (CIDEr 206.49, METEOR 23.29) with PDVC(ICCV2021') (CIDEr 162.08, METEOR 19.93) and HiCM2(AAAI2025')(CIDEr 113.88, METEOR 16.69), confirming that LOCO maintains competitive performance and generalization capabilities across different domains.
>
> **Real-world Motivation and Applications**
>
> HDVC technology is essential for enhancing user experience in OTT services and improving accessibility for the visually impaired. Real-world applications include Scene Navigation for OTT platforms (e.g., Netflix, Youtube), which provides episode summaries and timestamps/descriptions for key scenes.

---

### Official Review · Reviewer_WWFW · 2025-11-02

**Soundness:** 3
**Presentation:** 3
**Contribution:** 2
**Rating:** 4
**Confidence:** 3

**Summary:**

In this paper, the authors address a key limitation in Dense Video Captioning (DVC), i.e., the scale from short, minute-level clips to hour-long videos like movies. This failure is attributed to a lack of both appropriate benchmarks and models capable of handling extensive temporal dependencies and narrative context. To tackle this technical gap, the authors introduce a task, named Hierarchical Dense Video Captioning (HDVC), which requires both (i) scene-level dense captions with timestamps and (ii) a single video-level narrative paragraph for hour-long videos. To this end, HourHDVC, a new large-scale dataset, is proposed with comprehensive, hierarchical annotations for both scene-level and narrative-level captioning across hour-long video content. A new framework, LOCO (LOng COntext memory-based hierarchical dense video captioning), an end-to-end hierarchical DVC model, is also proposed.

**Strengths:**

1. The paper is, in general, well written in a good structure and is easy to follow.

2. The proposed problem is indeed a key challenge in dense video captioning, i.e., most prior DVC datasets target seconds–minute clips. HourHDVC fills an important gap (hour-scale, paragraph annotations, hierarchical structure). The dataset statistics and motivation are clearly presented.

3. The proposed LOCO model is well designed to directly tackle the new HDVC task. The two-tier memory system is a reasonable solution that explicitly maps to the problem's two-level (intra-scene and inter-scene) context, which the authors clearly define. The inclusion of a gradient flow mechanism across windows is a technically sound detail to improve learning over long sequences.

4. A new metric is proposed, which the authors claim is a better fit for the proposed task compared to traditional metrics(BLEU, CIDER). The proposed model has good performance in general. It is also good that lots of ablation tests are included.

**Weaknesses:**

1. The dataset is built upon the MAD-v2. This dataset is based on movie audio descriptions, which specifically focus on visual information for the visually impaired and intentionally do not overlap with dialogue. Hence, the "ground truth" annotations, even when processed by an LLM, are likely heavily biased toward visual actions rather than narrative points driven by dialogue, which is a critical part of movie narratives.

2. Although the authors include a user study for data quality, which compares LLM-generated captions to expert-refined LLM-generated captions, this is just to check the refinement step. It is still quite possible that the LLM pipeline introduces systematic biases in the benchmark already, which also indicates that the scalability and quality of the annotation pipeline are fundamentally limited by the accuracy, objectivity, and generalization of large language models, especially under challenging visual or audio ambiguity.

3. The proposed metric ConSimLlama3 uses an LLM evaluator (Llama3/GPT-4o style). While the human correlation numbers look good, LLM-based metrics can be unstable depending on model/version/prompt.

**Questions:**

Please check the "Weaknesses" section and the other questions as follows:

1. I wonder what exact LLM and temperature/chain-of-thought steps were used for ConSimLlama3? Can you provide an ablation where ConSim is computed with several models (e.g., Llama, GPT-style) to show metric robustness?

2. As mentioned, the proposed ConSimLlama3 applied an LLM. If LLM has pre-existing knowledge of the evaluation movies (which is quite possible), how can you be sure it isn't inflating the scores?

---

> ### Author Response · Authors · 2025-11-21
>
> We thank the reviewer for recognizing our motivation and LOCO's effectiveness. We will incorporate the feedback.
>
> **Speech Modality in Dataset Construction**
>
> As noted in Sec. 3.2, our method uses both Audio Descriptions and Speech Transcripts. Since transcripts contain full dialogue, this enables localization and captioning that consider important narrative context, rather than being visual-only grounded. Explicitly feeding dialogues forces the model to align visual cues with spoken narrative, mitigating the visual-bias in raw audio descriptions. As shown in Fig. 16 (Supp.), our captions capture high-level semantics like "leaving the Avengers in mourning" or "symbolically ending one era," which cannot be derived from simple visual descriptions.
>
> **Quality and Scalability of LLM-based Dataset Construction Pipeline**
>
> Quantitative and qualitative analyses indicate that noise in our LLM-based annotations is minimal. We compared the pre-refinement (GPT-generated) and post-refinement (human-refined) evaluation sets for 364 scenes and observed an IoU of 0.80 between the two timestamp sets, indicating that the raw LLM outputs already provide accurate scene localization with only minor boundary differences. On the textual side, experts edited only 33 of 364 captions (≈9%), and the hallucination rate was ≈1.9%, showing that systematic LLM artifacts are rare in practice, even under visually or acoustically ambiguous conditions.
>
> To rigorously address the concern of single-model bias, we conducted a cross-model validation using Gemini 2.5 Pro and GPT-5. For scenes with high temporal alignment (IoU $>$ 0.7), we compared their outputs against our GPT-4o results that used in our data generation pipeline. We compared their generated captions against our GPT-4o results for scenes with high temporal alignment (IoU > 0.7). The results showed low lexical overlap (Average BLEU-4: 7.90 vs. Gemini, 5.20 vs. GPT-5) but high semantic agreement (ConSim: 89.80 vs. Gemini, 87.40 vs. GPT-5.1). This combination of diverse wording (low BLEU) and identical meaning (high ConSim) confirms that our pipeline extracts objective narrative facts grounded in the source inputs (Transcript/AD), rather than reflecting the stylistic bias of a single model.
>
> Consistently, Table 6 shows that a model trained on noisy (pre-refinement) annotations achieves Recall 29.58, which is very close to 32.54 when trained on fully refined data; while refinement brings small gains, this gap suggests that residual LLM noise does not undermine the validity of the benchmark. Also, in the Quality Control (Sec 6.1), we evaluated 20 samples each before(LLM-generated) and after refinement(Human-refined) using 7 evaluators on a 10-point scale. We clarify that we evaluated the pre- and post-refinement samples separately (not a comparative relative evaluation) and did not provide any meta-information to the participants. The result—8.49 for GPT-generated captions and 8.51 for expert-refined captions—demonstrates that the raw GPT-generated data already possesses sufficiently high quality. This statistical parity implies that our pipeline is competitive in terms of scalability, as it ensures high-quality annotations without expensive human refinement.
>
> **Robustness of ConSim Metric**
>
> In response to the feedback, we verified the robustness of the ConSim metric by checking its alignment with human judgments using various LLM models and versions. To isolate the effect of the evaluator itself, we used the exact same evaluation prompt and instructions across all LLMs. As shown in Section 6.3 Human Alignment of ConSim, we measured the Pearson correlation between evaluation scores from 10 field experts and automatic metrics for randomly selected 7 scene predictions and 3 narrative predictions. We utilized Llama3 70B, Llama3.3 70B, and ChatGPT-5.1 Instant. All three models showed the highest correlation with human judgment (METEOR: 0.27, ROUGE_L: 0.07, ConSim_Llama3: 0.87, ConSim_Llama3.3: 0.75, ConSim_Gpt5.1: 0.84). Importantly, even with the same prompt, these different models and versions produced highly consistent correlations with human judgments, indicating that ConSim is not sensitive to evaluator choice. Both Llama3.3 70B (a different version of the same series) and ChatGPT-5.1 Instant (a larger model) demonstrated similarity to human judgment when following our proposed instructions, thereby validating the ConSim metric. For Llama3 and Llama3.3, we set temperature=0.6 and top-p=0.9. Note that we measured ConSim three times in total and used the mean and standard deviation values to ensure the stability of the evaluation.

---

> > ### Author Response · Authors · 2025-11-21
> >
> > **Pre-existing Knowledge and ConSim Scores**
> >
> > The proposed ConSim measures the semantic similarity between two paragraphs, so it does not suffer from inflation issues. We demonstrated that ConSim is not subject to exaggeration effects due to LLM knowledge. For the 10 samples evaluated in Section 6.3  Human Alignment of ConSim, humans gave an average score of 3.84, while Llama3 70B gave 3.21, Llama3.3 70B gave 3.10, and ChatGPT-5.1 Instant gave 3.30. The LLM-based scores are obtained by averaging over three independent runs for each model, whereas the human score is the average over 10 annotators. This verifies that LLM-based methods do not suffer from inflation; rather, they utilize rich prior knowledge to apply rigorous evaluation criteria similar to (or stricter than) humans. Thus, the performance reported in our paper is reliable and not inflated.

---

### Author Response · Authors · 2025-12-03
**General Response to Area Chair and Reviwers [1/2]**

**Dear Area Chair**,

We sincerely appreciate the Area Chair’s time and thoughtful handling of our submission under these unusual circumstances.

In brief, our work identifies a core limitation of existing Dense Video Captioning methods: their inability to scale from minute-level clips to hour-long videos due to the absence of suitable benchmarks and models for long-range dependency reasoning. To address this gap, we introduce Hierarchical Dense Video Captioning (HDVC) and the accompanying HourHDVC dataset, the first large-scale benchmark providing both scene-level annotations and a global narrative paragraph for hour-long videos. We further propose LOCO, a two-tier memory architecture designed to model hierarchical temporal structure, yielding strong performance on both scene-level and narrative-level captioning. Our evaluation also leverages the ConSim metric, which provides a context-sensitive similarity measure with strong human alignment.

We thank the all reviewers for their constructive feedback. Based on the reviews, we have made significant improvements to the manuscript.

---

## Paper Updates
Following the reviewers' valuable suggestions, we have substantially revised our manuscript and added several new experiments and analyses. All changes are highlighted in **blue** in the revised version.
### **A. Clarifications and Enhancements to Existing Contents**

-   **Clarification of Evaluation Detail and Notation**
    Table 2 now clearly marks zero-shot baselines, and Equation (2) includes explicit symbol definitions for readability.

-    **Human Refinement Process**

     Sections 3.2 and 6.1 were expanded to describe how human annotators refined timestamps and validated text quality.

-   **Generalization on Ego4D-HCap**
    Table 5 now includes HiCM2 and PDVC for a more complete generalization comparison.
### **B. New Analyses and Newly Added Contents**

-   **Potential LLM Biases and Dataset Errors**
    Section 6.1 now includes new quantitative analysis for LLM-generated refinement reliability and single-model bias.

-   **Robustness of ConSim Metric**
    Section 6.3 adds multi-model evaluations analysis confirming ConSim’s robustness across LLMs.

-   **Compute Efficiency of LOCO**
    A new analysis in Section 6.2 reports inference/runtime comparisons showing LOCO’s minimal overhead over Vid2Seq-SW.

-   **Window Size Analysis**
    Supplementary D introduces a four-window comparison motivating the choice of 600 s.

-   **Sensitivity to ASR Transcript Quality**
    Section 5.4 and Figure 3 include a transcript-drop robustness experiment analysis.

-   **Robustness to Noisy Speech**
    Supplementary D adds an analysis of LOCO’s behavior under noisy speech.
---
## Summary of Strengths Highlighted by Reviewers

### **■ Clear Motivation & Dataset Contribution (HDVC + HourHDVC)**

• "*The proposed problem is indeed a key challenge ... HourHDVC fills an important gap (hour-scale, paragraph annotations, hierarchical structure). The dataset statistics and motivation are clearly presented.*" **~Reviewer WWFW**

• "*The authors introduce a new task and dataset ... This dataset is likely to be valuable for future research on long-form video understanding.*" **~Reviewer 1GSZ**

• "*It introduces a novel and well-defined task (HDVC) supported by the first large-scale, richly annotated dataset (HourHDVC) for hour-long videos.*" **~Reviewer jmSY**

### **■ Model Contribution & Technical Innovation (LOCO)**

• "*The proposed LOCO model is technically innovative, featuring an elegant two-tier memory design that effectively handles long-range dependencies, which is validated through extensive ablations.*" **~Reviewer jmSY**

• "*The proposed LOCO model is well designed to directly tackle the new HDVC task. The two-tier memory system is a reasonable solution ... a gradient flow mechanism across windows is a technically sound detail to improve learning over long sequences.*" **~Reviewer WWFW**


### **■ Experiments**

• "*... which is validated through extensive ablations ... The claims are further solidified by thorough evaluation, including strong baseline comparisons and a careful validation ...*" **~Reviewer jmSY**

• "*The experiments are comprehensive for the proposed benchmark, including ablations and human evaluations.*" **~Reviewer AHWX**

• "*The proposed model has good performance in general. It is also good that lots of ablation tests are included.*" **~Reviewer WWFW**

### **■ Evaluation Metric (ConSim)**

• "*The claims are further solidified by thorough evaluation, ... a careful validation of the proposed ConSim metric.*" **~Reviewer jmSY**

• "*A new metric is proposed, which the authors claim is a better fit for the proposed task compared to traditional metrics.*" **~Reviewer WWFW**

• "*They also propose a new evaluation metric, ConSim, designed to measure how well models capture contextual information.*" **~Reviewer 1GSZ**

---

> ### Author Response · Authors · 2025-12-03
> **General Response to Area Chair and Reviwers [2/2]**
>
> ## Resolution of Reviewer Comments
>
> | Key Comments | Resolution (Rebuttal Actions) | Reviewers |
> |------------|--------------------------------|-----------|
> | Data Quality and Bias Verification | **Resolved**: Human edits required for only 9% of captions; hallucination rate ≈1.9%; quality scores for GPT-only vs. refined captions are nearly identical (8.49 vs. 8.51). Also, Cross-model validation (GPT-4o vs. Gemini 2.5 Pro vs. GPT-5.1) shows high semantic agreement despite low lexical overlap, indicating absence of single-model bias. | WWFW, AHWX, jmSY |
> | Metric stability across evaluator LLMs | **Resolved**: ConSim exhibits strong and consistent human correlation across Llama3/Llama3.3/ChatGPT-5.1 (0.75–0.87) under identical prompts (Sec. 6.3). | WWFW |
> | Clarification of HDVC | **Resolved**: We clarified that GPT-4o is used only for text-to-text data construction, whereas HDVC requires autonomous video-to-caption reasoning over hour-long inputs, which current LLM/Video-LLM pipelines fail to handle (localization, long-term context, narrative consistency). | AHWX |
> | Clarification of simple clip-wise pipelines | **Resolved**: Empirical results (Table 2) show that sliding-window and clip-merge baselines fail substantially (METEOR ~1.28, F1 <30.39) compared to LOCO; these pipelines cannot model long-range dependencies across hours. | AHWX, jmSY |
> | Effect of Context-aware Memory on Scene DVC | **Resolved**: We showed the drop stems from multi-task optimization trade-offs; when isolating Scene DVC training, Context-aware Memory clearly improves performance (METEOR 6.51 vs. 6.32, F1 45.08 vs. 43.96). We also showed complementary roles of context-aware memory and long-term memory.  | 1GSZ |
> | Computational cost and scalability of LOCO | **Resolved**: LOCO requires only a single 24GB GPU, uses pre-extracted CLIP features, and processes an hour-long video in ~36 seconds, confirming practical scalability. | 1GSZ, jmSY |
> | Robustness to noisy speech and transcription quality | **Resolved**: Evaluation on Les Misérables vs. Signs shows stable performance even with heavy background music; LOCO leverages strong visual grounding to mitigate degraded speech. Also, controlled transcript-drop experiments show stable performance up to 30% loss, confirming graceful degradation and robustness to imperfect ASR.| jmSY |
>
> ***
>
> ## Additional Message to Area Chair
>
> We sincerely thank the Area Chair and reviewers for the time and thoughtful care devoted to handling our submission throughout the review and rebuttal processes.
>
> During the rebuttal period, we had extensive and constructive discussions with all reviewers. We addressed their comments with additional analyses, including cross-model ConSim validation, efficiency analysis, and detailed clarifications on proposed data. These additions helped strengthen both the empirical rigor and interpretability of the revised manuscript.
>
> We observe that prior Dense Video Captioning research has been restricted to minute-level clips, with no formal task addressing the consistent understanding of scene transitions, contextual flow, and narrative progression in videos longer than an hour. To fill this gap, we define HDVC and introduce HourHDVC, a benchmark built on ~2-hour videos with both scene-level events and narrative paragraphs. This dataset enables systematic analysis of long-range dependencies, contextual continuity, and narrative structure beyond what clip-based approaches can capture, and its reliability is supported by extensive refinement and validation.
>
> We believe these contributions provide a solid foundation for future research on hour-long video understanding and offer a practical basis for more efficient automatic highlight event detection, timestamp alignment, and contextual description generation in real-world hour-long videos, introducing the first task and benchmark that enable systematic study of hierarchical Dense Video Captioning in hour-long videos.

---

### Meta-Review · Area_Chair_1GtA · 2025-12-25

**Summary:**

The paper proposes a hierarchical dense video captioning task for long-form videos and introduces a new dataset HourHDVC and the LOCO model.

The initial reviewer scores are 4, 2, 4, and 10. Across the reviews, several major concerns were consistently raised:
- Dataset quality and potential biases introduced by the use of LLMs in data construction (WWFW, AHWX, jmSY).
- Stability of the LLM-based evaluation metrics  (WWFW, jmSY).
- Lack of strong baseline comparisons, e.g., clip-based captioning + summarization using SOTA MLLMs  (AHWX, jmSY).
- Insufficient technical details (1GSZ).
- Generalization and more compared methods on other domains (AHWX, 1GSZ, jmSY)

In the rebuttal, the authors partially addressed concerns regarding dataset construction and evaluation, but the response to the baseline comparison issue was not fully satisfactory. In addition, validation on domains beyond popular movies is encouraged to help address concerns about potential data leakage.

Due to limitations in baseline coverage and presentation quality, the AC recommends rejecting the paper. The authors are encouraged to address these issues and resubmit to a next venue.

**Reviewer Concerns:**

See above.

**Reviewer Scores:**

Despite substantial divergence in the initial scores, the reviewers are largely aligned in their primary concerns which are not fully addressed.

---

### Decision · Program_Chairs · 2026-01-26

Reject